# Controllable and Compositional Generation with Latent-Space Energy-Based Models

**Weili Nie**
NVIDIA
wnie@nvidia.com

**Arash Vahdat**
NVIDIA
avahdat@nvidia.com

**Anima Anandkumar**
Caltech, NVIDIA
anima@caltech.edu

## Abstract

Controllable generation is one of the key requirements for successful adoption of deep generative models in real-world applications, but it still remains as a great challenge. In particular, the compositional ability to generate novel concept combinations is out of reach for most current models. In this work, we use energy-based models (EBMs) to handle compositional generation over a set of attributes. To make them scalable to high-resolution image generation, we introduce an EBM in the latent space of a pre-trained generative model such as StyleGAN. We propose a novel EBM formulation representing the joint distribution of data and attributes together, and we show how sampling from it is formulated as solving an ordinary differential equation (ODE). Given a pre-trained generator, all we need for controllable generation is to train an attribute classifier. Sampling with ODEs is done efficiently in the latent space and is robust to hyperparameters. Thus, our method is simple, fast to train, and efficient to sample. Experimental results show that our method outperforms the state-of-the-art in both conditional sampling and sequential editing. In compositional generation, our method excels at zero-shot generation of unseen attribute combinations. Also, by composing energy functions with logical operators, this work is the first to achieve such compositionality in generating photo-realistic images of resolution $1024\times1024$.

## 1 Introduction

Deep generative learning has made tremendous progress in image synthesis. High-quality and photorealistic images can now be generated with generative adversarial networks (GANs) [17, 28, 29], score-based models [23, 51], and variational autoencoders (VAEs) [32, 54]. However, a key requirement for the success of these generative models in many real-world applications is the controllability of the generation process. Controllable generation, in particular the compositional ability to generate novel concept combinations, still remains a great challenge to current methods.

A common approach for achieving controllable generation is to train a *conditional* model [36, 31, 10, 41, 38], where the conditional information, such as semantic attributes, is specified during training to guide the generation. There are two major drawbacks with this approach: (i) Since the set of attributes is fixed and pre-defined, it is difficult to introduce new attributes to an existing model. This, in turn, limits the model's compositional ability for unseen attributes and their novel combinations [12]. (ii) Training conditional generative models from scratch for new attributes is computationally expensive, and it can be challenging to reach to the same image quality of the uncontrollable counterparts [10, 41].

Recent works have attempted to overcome these issues by first training an unconditional generator, and then converting it to a conditional model with a small cost [26, 20, 1, 42]. This is often achieved by discovering semantically meaningful directions in the latent space of the unconditional model. This way, one would pay the most computational cost only once for training the unconditional model. However, these approaches often struggle with compositional generation, in particular with rare

35th Conference on Neural Information Processing Systems (NeurIPS 2021).

combination of the attributes and the introduction of new attributes. An appealing solution to the compositionality problem is to use energy-based models (EBMs) for controllable generation [12, 18, 13]. This is due to the fact that energy functions representing different semantics can be combined together to form compositional image generators. However, existing approaches train EBMs directly in the pixel space, making them slow to sample and difficult to generate high-resolution images [13].

In this paper, we leverage the compositionality of EBMs and the generative power of state-of-the-art pre-trained models such as StyleGANs [28, 29] to achieve high-quality controllable and compositional generation. Particularly, we propose an EBM that addresses the problem of controlling an existing generative model instead of generating images directly with EBMs or improving the sampling quality.

**Our main contributions are summarized as follows**:

- We propose a novel formulation of a joint EBM in the latent space of pre-trained generative models for controllable generation. In particular, the EBM training reduces to training a classifier only, and it can be applied to any latent-variable generative model.
- Based on our EBM formulation, we propose a new sampling method that relies on ordinary different equation (ODE) solvers. The sample quality is robust to hyperparameters.
- We show that our method is fast to train and efficient to sample, and it outperforms previous methods in both conditional sampling and sequential editing.
- We show strong compositionality with zero-shot generation on unseen attribute combinaions, and with logical compositions of energy functions in generating high-quality images.

Specifically, we build an EBM in the joint space of data and attributes where the marginal data distribution is denoted by an implicit distribution (e.g., a pre-trained GAN generator), and the conditional distribution of attributes, given data, is represented by an attribute classifier. Using the reparameterization trick, we show that our EBM formulation becomes an energy function in the latent space where the latent distribution is known (i.e., a standard Gaussian). This way, we only need to train the classifier in the data space and do the sampling in the latent space. Because our method only requires training a classifier to add controllability, it is conceptually simple and fast-to-train.

For sampling, most existing EBMs rely on the Langevin dynamics (LD) sampling which is often computationally expensive and sensitive to the choice of sampling parameters. Instead, we build on the recent score-based model [51], and define the sampling process in the latent space using the corresponding probability flow ODEs, induced by the reverse diffusion process. Thus, we rely on an ODE solver for sampling from our EBM formulation in the latent space. With its adaptive step size, the ODE sampling in the latent space is efficient and robust to the sampling parameters.

In experiments, we show our method achieves higher image quality and better controllability in both the conditional sampling and sequential editing, compared to various baselines, including StyleFlow [1] and JEM [18]. For the training time, we show our method is $25\times$ faster than StyleFlow on the FFHQ data. For the inference time, our sampling is at least $49\times$ and $876\times$ faster than EBMs [18, 12] and score-based models [51] in the pixel space, respectively, on CIFAR-10. More importantly, our method excels on zero-shot generation with unseen attribute combinations where StyleFlow almost completely fails. By composing energy functions with logical operators [12], our method is the first to show such compositionality in generating photo-realistic images of resolution $1024\times1024$.

## 2 Method

In this section, we will first give an overview of EBMs and then propose our new EBM formulation in the latent space, based on which, we introduce a new sampling method through the ODE solver.

### 2.1 Energy-based models

EBMs [34] represent data $x \in \mathbb{R}^d$ by learning an unnormalized probability distribution $p_\theta(x) \propto e^{-E_\theta(x)}$, where $E_\theta(x) : \mathbb{R}^d \to \mathbb{R}$ is the energy function, parameterized by a neural network. To train an EBM on a data distribution $p_{\text{data}}$ with maximum likelihood estimation (MLE), we can estimate the gradient of the data log-likelihood $L(\theta) = \mathbb{E}_{x \sim p_{\text{data}}}[\log p_\theta(x)]$ as [22]:

$$\nabla L(\theta) = \mathbb{E}_{x \sim p_{\text{data}}}[\nabla_\theta E_\theta(x)] - \mathbb{E}_{x \sim p_\theta}[\nabla_\theta E_\theta(x)] \tag{1}$$

To sample $x$ from $p_\theta$ for training and inference, Langevin dynamics (LD) [57] is applied as follows,

$$x_0 \sim p_0(x), \quad x_{t+1} = x_t - \frac{\eta}{2}\nabla_x E_\theta(x_t) + \epsilon_t, \quad \epsilon_t \sim N(0, \eta I) \tag{2}$$

where $p_0$ is the initial distribution, $\eta$ is the step size. When $\eta \to 0$ and $t \to \infty$, $x_t$ is guaranteed to follow the probability $p_\theta(x)$ under some regularity conditions [50].

## 2.2 Modelling joint EBM in the latent space

To formulate our EBM for controllable generation, we consider the following setting: We have the data $x \in \mathcal{X} \subseteq \mathbb{R}^d$ and its attribute code $c = \{c_1, \cdots, c_n\} \in \mathcal{C} \subseteq \mathbb{R}^n$, which is a $n$-dimensional vector with each entry $c_i$ representing either a discrete $m_i$-class attribute where $c_i \in \{0, \cdots, m_i - 1\}$, or a continuous attribute where $c_i \in \mathbb{R}$. The goal is to learn a generative model and semantically control its generated samples by manipulating its conditioning attribute code $c$. To begin with, we define the joint generative model on both the data and attribute as:

$$p_\theta(x, c) := p_g(x)p_\theta(c|x) \propto p_g(x)e^{-E_\theta(c|x)} \tag{3}$$

where $p_g(x)$ is an implicit distribution defined by a pre-trained generator $g$ (such as GANs) in the form $x = g(z)$ with the latent variable $z \in \mathcal{Z}$. And, $p_\theta(c|x)$ is conditional distribution on attributes given $x$, modeled by a conditional energy function $E_\theta(c|x)$.

In this paper, we assume that the generator $g$ is fixed and we only train the energy function $E_\theta(c|x)$. We also assume the attributes are conditionally independent, i.e., $E_\theta(c|x) = \sum_{i=1}^n E_\theta(c_i|x)$. Since, our goal is to preserve the image generation quality, we design the joint in Eq. (3) such that the marginal data distribution $p_\theta(x)$ from our joint satisfies $p_\theta(x) = p_g(x)$. This is obtained with an energy function that is normalized up to a constant. Thus, we define:

$$E_\theta(c_i|x) = \begin{cases} -f_i(x; \theta)[c_i] + \log \sum_{c_i} \exp\left(f_i(x;\theta)[c_i]\right) & \text{if } c_i \text{ is discrete,} \\ \frac{1}{2\sigma^2}(c_i - f_i(x;\theta))^2 & \text{if } c_i \text{ is continuous} \end{cases} \tag{4}$$

where $f_i(x; \theta)$ is the output of a multi-class classifier mapping from $\mathcal{X}$ to $\mathbb{R}^{m_i}$ if the $i$-th attribute is discrete or a regression network mapping from $\mathcal{X}$ to $\mathbb{R}$ if it is continuous. Without loss of generality, we will always call $f_i(x; \theta)$ a classifier. Here, $\sigma^2$ is a hyperparameter to adjust the continuous energy. In practice, we set $\sigma^2 = 0.01$ after normalizing all the continuous attributes to $[0, 1]$.

We cannot use Eq. (3) for sampling, since $p_g(x)$ is defined implicitly in GAN generators. However, in Appendix A.1, we show that using the reparameterization trick [6, 58], sampling $x$ from $p_\theta(x, c)$ is reparameterized to sampling $z$ from $p_\theta(z, c) \propto e^{-E_\theta(c|g(z)) + \log p(z)}$, and then transferring the $z$ samples to the data space using generator $x = g(z)$. In most generative models, the prior distribution $p(z)$ is a standard Gaussian. Thus, the joint becomes $p_\theta(z, c) \propto e^{-E_\theta(z, c)}$ with the energy function:

$$E_\theta(z, c) = \sum_i E_\theta(c_i|g(z)) + \frac{1}{2}\|z\|_2^2 \tag{5}$$

where each conditional energy function $E_\theta(c_i|g(z))$ is given by Eq. (4). For sampling, we can run LD of Eq. (2) in the $z$-space using the energy function above. At the end of a chain, we pass the final $z$ samples to the generator $g$ to get the $x$ samples. Since our latent-space EBM is formulated for controllable generation, we term it as **LA**tent **C**ontrollable **E**BM (LACE) throughout this work.

Inspecting the joint energy $E_\theta(z, c)$ in Eq. (5) shows that when the unconditional generator $g$ is fixed, the only trainable component is the classifier $f_i(x; \theta)$ that represents $E_\theta(c_i|x)$. Therefore, unlike the previous joint EBMs [18] that train both $p_\theta(x)$ and $p_\theta(c|x)$, our method only needs to train a classifier for the conditional $p_\theta(c|x)$, and does not require sampling with LD in the $x$-space during training. That is, for controllable generation, we can train the classifier only in the $x$-space and do the LD sampling in the $z$-space. This fact makes our method conceptually simple and easy to train.

Note that the considered $x$-space for training the classifier does not have to be the pixel space. Instead, we can choose any intermediate layer of the pre-trained generator as the $x$-space. Take StyleGAN2 [29] as an example, if we train the classifier in the $w$-space, then $g(z)$ in Eq. (5) actually corresponds to the mapping network of StyleGAN2, and once we obtain the $w$ samples via the LD sampler, we can pass them to the synthesis network of StyleGAN2 to get the final images. In other words, since data $x$ is a deterministic function of $z$, the distribution of attributes is uniquely defined by $z$. We have the freedom to define $p_\theta(c_i|z)$ by applying the classifier on top of $z$ or any representation extracted from $z$ including the generator $g(z)$ or its intermediate representations such as $w$-space in StyleGAN2.

## 2.3 Sampling through an ODE solver

Sampling with LD requires hand-crafted tricks to speed up and stabilize its convergence [14, 18]. Even in the latent space, our experiments show that LD tends to be sensitive to its hyperparameters. In

this section, we introduce another sampling method, called *ODE sampler*, that relies on an ODE solver. Our method with the LD and ODE sampler is termed as *LACE-LD* and *LACE-ODE*, respectively.

Our idea is inspired by the prior work [51], which shows that controllable generation can be achieved by solving a conditional reverse-time SDE. Specifically, if we consider a *Variance Preserving (VP) SDE* [51], the forward SDE is defined as $dx = -\frac{1}{2}\beta(t)xdt + \sqrt{\beta(t)}dw$, where $w$ is a standard Wiener process and $x_0 \sim p_{\text{data}}$ is a data sample. The scalar time-variant diffusion coefficient $\beta(t)$ has a linear form $\beta(t) = \beta_{\min} + (\beta_{\max} - \beta_{\min})t$, where $t \in [0, T]$. Then, the conditional sampling from $p_0(x|c)$ is equivalent to solving the following reverse VP-SDE:

$$dx = -\frac{1}{2}\beta(t)[x + 2\nabla_x \log p_t(x, c)]dt + \sqrt{\beta(t)}d\bar{w} \qquad (6)$$

where $\bar{w}$ is a standard reverse Wiener process when time flows backwards from $T$ to 0, and $p_t(x, c)$ is the join data and attribute distribution at time $t$. Song et al. [51] show that there exist a corresponding ODE for the SDE in Eq. (6) which can be used for sampling. In Appendix A.2, using the reparameterization trick, we show that the ODE in the latent space for our model takes a simple form:

$$dz = \frac{1}{2}\beta(t) \sum_i \nabla_z E(c_i|g(z))dt \qquad (7)$$

with negative time increments from $T$ to 0. To generate a sample $x$ conditioned on $c$, all we need is to draw $z(T) \sim N(0, I)$ and solve the ODE in Eq. (7) using a black-box solver. The final $z(0)$ samples are transferred to the data space through the generator $x = g(z(0))$. Our main insight in Appendix A.2 that leads to the simple ODE is that the latent $z$ in most generative models follows a standard Gaussian distribution, and diffusing it with VP-SDE will not change its distribution, i.e., $p_t(z(t)) = \mathcal{N}(0, I)$.

**Remarks** There are some key observations from our ODE formulation in Eq. (7):

*(i) Connections to gradient flows.* Similar to [45, 3] that build connections between SDE/ODE and gradient flows, our ODE sampler can be seen as a gradient flow that refines a latent $z$ from a random noise to a $z$ vector conditioned on an attribute vector. While [3] relied on the Euler-Maruyama solution of the SDE, we convert our generative SDE to an ODE. Adaptive discretization with a higher order method (e.g., Runge-Kutta [7]) is often preferred when solving ODEs. But our ODE formulation also works well with the first-order Euler method, and its performance lies in-between LACE-LD and LACE-ODE (See Appendix A.11.3).

*(ii) Advantages of ODE in the latent space.* If the SDE/ODE is built in the pixel space as in [51], it requires 1) estimating the score function $\nabla_x \log p_t(x(t))$, and 2) training a *time-variant* classifier $p_t(c|x(t))$, both of which make its training and inference challenging. Instead, our ODE sampler is much simpler: we do not need to train any score function. We only train a time-invariant classifier. Compared with the LD sampler that uses a fixed step size and is sensitive to many hyperparameters (e.g., step size, noise scale and number of steps), our ODE sampler is adaptive in step sizes and only needs to tune the tolerances, making it more efficient and robust to hyperparameters.

## 3 Experiments

In this section, we show the effectiveness of our method in conditional sampling, sequential editing and compositional generation, and we also perform an ablation study on the sampling method.

**Experimental setting** We use StyleGAN-ADA [27] as the pre-trained model for experiments on CIFAR-10 [33], and StyleGAN2 [29] for experiments on FFHQ [28]. We train the classifier in the $w$-space, where our method works best (see ablation studies in Appendix A.5). To train the latent classifier in the $w$-space, we first generate (image, $w$) pairs from StyleGAN, and then label each $w$ latent by annotating its paired image with an image classifier (see data preparation in Appendix A.3).

For LACE-ODE, we use the 'dopri5' solver [7] with the tolerances of (1e-3, 1e-3), and we set $T = 1$, $\beta_{\min} = 0.1$ and $\beta_{\max} = 20$. For LACE-LD, the step size $\eta$ and the standard deviation $\sigma$ of $\epsilon_t$ are chosen separately for faster training [18], where the number of steps $N = 100$, step size $\eta = 0.01$ and standard deviation $\sigma = 0.01$. For metrics, we use (i) *conditional accuracy* (*ACC*) to measure the controllability, where we generate images using randomly sampled attribute codes, and pass them to a pre-trained image classifier to predict the attribute codes, and (ii) *FID* to measure the generation quality and diversity [21]. See Appendix A.3 for more details.

Table 1: Comparison of our method and baselines for conditional sampling on CIFAR-10. For notations, Train – training time, Infer – inference time (m: minute, s: second), which refer to the single GPU time for generating a batch of 64 images, $\eta$ is the LD step size, and $N$ is the number of predictor steps in the PC sampler.

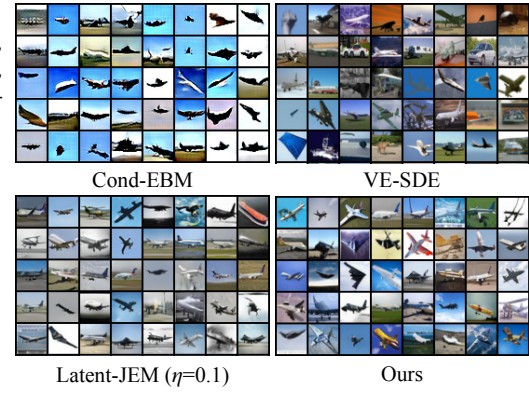

| Methods | Train | Infer | FID↓ | ACC↑ |
|---|---|---|---|---|
| JEM [18] | 2160m | 135s | 52.35 | 0.645 |
| Cond-EBM [12] | 2280m | 24.5s | 41.72 | 0.792 |
| VP-SDE [51] | 52800m | 438s | 19.13 | 0.643 |
| VE-SDE [51] | 52800m | 448s | **2.97** | 0.662 |
| Latent-JEM ($\eta$=0.1) | 21m | 0.63s | 8.75 | 0.950 |
| Latent-JEM ($\eta$=0.01) | 21m | 0.63s | 5.65 | 0.821 |
| LACE-PC ($N$=100) | **4m** | 0.84s | **2.99** | 0.747 |
| LACE-PC ($N$=200) | **4m** | 1.86s | **2.94** | 0.722 |
| LACE-LD | **4m** | 0.68s | 4.30 | 0.939 |
| LACE-ODE | **4m** | **0.50s** | 6.63 | **0.972** |

Cond-EBM        VE-SDE

Latent-JEM ($\eta$=0.1)        Ours

Figure 1: Conditionally generated images of our method (LACE-ODE) and baselines on the plane class of CIFAR-10. See Appendix A.6 for more results.

### 3.1 Conditional sampling

**Baselines** For comparison, we consider a set of baselines: StyleFlow [1], JEM [18], Conditional EBM (Cond-EBM) [12], and SDEs [51]. We also propose *Latent-JEM*, a variant of JEM modelled in the latent space, and *LACE-PC*, which replaces the ODE sampler with the Predictor-Corrector (PC) sampler that solves the reverse SDE (Eq. 6) [51]. See the Appendix A.4 for more details.

**CIFAR-10** The results of our method against baselines on CIFAR-10 are shown in Table 1 and Figure 1. Our method requires only four minutes to train, and it takes less than one second to sample a batch of 64 images on a single NVIDIA V100 GPU, which significantly outperforms previous EBMs (at least 49× faster) and score-based models (at least 876× faster) in the pixel space. Latent-JEM that works in the latent space is also slower in training as it performs the LD for each parameter update.

For the conditional sampling performance, LACE-ODE and LACE-LD largely outperform all the baselines in precisely controlling the generation while maintaining the relatively high image quality. In particular, LACE-PC performs similarly to VE-SDE: they can achieve better image quality but have a problem with precisely controlling the generation (ACC≤0.75). Besides, we observe that EBMs in the latent space achieve much better overall performance than EBMs in the pixel space.

**FFHQ** To test on FFHQ, we use 10k $(w, c)$ pairs created by [1] for training, where $w$ is sampled from the $w$-space of StyleGAN2, and $c$ is the attribute code. Unless otherwise specified, we use truncation $\psi = 0.7$ for our method. Following the evaluation protocol from [1], we use 1k generated samples from StyleGAN2 to compute the FID. Note that given the small sample size, FID values tend to be high. For the reference, the original unconditional StyleGAN2 with 1k samples has FID 20.87.

The results of comparing our method with baselines are shown in Table 2 and Figure 2, where we condition on `glasses` and `gender_smile_age`, respectively. For the training time, our method only takes 2 minutes, which is around 25× faster than StyleFlow, and 5× faster than Latent-JEM. Our inference time increases with the number of attributes to control. For instance, LACE-ODE needs similar inference time with StyleFlow (0.68s vs. 0.61s) on `glasses`, but more inference time than StyleFlow (4.81s vs. 0.61s) on `gender_smile_age`. In practice, we could adjust the ODE tolerances, allowing for trade-offs between inference time and overall performance. Besides, we can also optimize the network to further reduce the inference time (see results in Appendix A.7).

For the controllability, both LACE-ODE and LACE-LD outperform the baselines by a large margin. Also, the generation quality of our method is on par with our proposed baselines, and much better than the prior work StyleFlow. For instance, LACE-ODE has much lower FID than StyleFlow (24.52 vs. 43.55). Latent-JEM achieves better FID but always has worse controllability. These quantitative results could be verified by the visual samples in Figure 2, where our method achieves high-quality controllable generation. However, StyleFlow has difficulty with (i) the full controllability in different cases, and (ii) the lack of image diversity specifically when conditioning on more attributes.

### 3.2 Sequential editing

In sequential editing, we semantically edit the images by changing an attribute each time without affecting other attributes and face identity. Given a sequence of attributes $\{c_1, \cdots, c_n\}$, we adapt our method for sequential editing by relying on the compositionality of energy functions. We define the

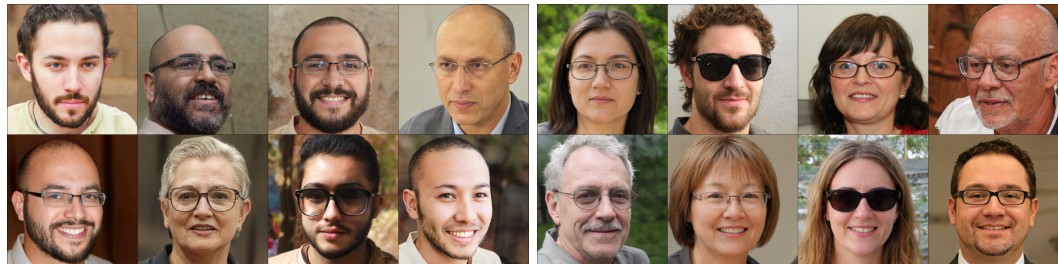

(a) `glasses=1` (*Left*: StyleFlow, *Right*: Ours)

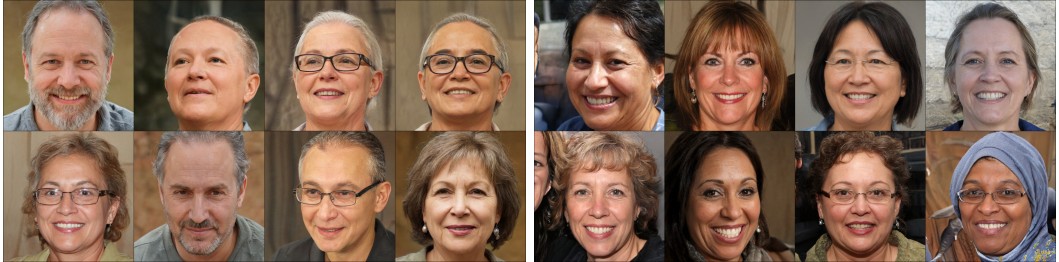

(b) `gender=female,smile=1,age=55` (*Left*: StyleFlow, *Right*: Ours)

Figure 2: Conditionally generated images of our method (LACE-ODE) and StyleFlow [1] on the `glasses` and `gender_smile_age` of FFHQ, respectively. Visually, our method achieves higher image generation quality, more diversity and better controllability than StyleFlow.

Table 2: Comparison between our method and baselines for conditional sampling on the `glasses` and `gender_smile_age` of FFHQ, respectively. For notations, Train – training time, Infer – inference time (m: minute, s: second), which refer to the single GPU time for generating a batch of 16 images, $\eta$ is the LD step size, and $N$ is the number of predictor steps in the PC sampler.

| Methods | Train | glasses | | | gender_smile_age | | | | |
| --- | --- | --- | --- | --- | --- | --- | --- | --- | --- |
| | | Infer | FID↓ | ACC$_{gl}$↑ | Infer | FID↓ | ACC$_{ge}$↑ | ACC$_{s}$↑ | ACC$_{a}$↑ |
| StyleFlow [1] | 50m | **0.61s** | 42.08 | 0.899 | **0.61s** | 43.88 | 0.718 | 0.870 | 0.874 |
| Latent-JEM ($\eta$=0.1) | 15m | 0.69s | 22.83 | 0.765 | 0.93s | 22.74 | 0.878 | 0.953 | 0.843 |
| Latent-JEM ($\eta$=0.01) | 15m | 0.69s | 21.58 | 0.750 | 0.93s | **21.98** | 0.755 | 0.946 | 0.831 |
| LACE-PC ($N$=100) | **2m** | 1.29s | 21.48 | 0.943 | 2.65s | 24.31 | 0.951 | 0.922 | 0.896 |
| LACE-PC ($N$=200) | **2m** | 2.20s | 21.38 | 0.925 | 4.62s | 23.86 | 0.949 | 0.914 | 0.894 |
| LACE-LD | **2m** | 1.15s | **20.92** | **0.998** | 2.40s | 22.97 | 0.955 | 0.960 | **0.913** |
| LACE-ODE | **2m** | 0.68s | **20.93** | **0.998** | 4.81s | 24.52 | **0.969** | **0.982** | **0.914** |

joint energy function of the $i$-th edit as

$$E_\theta^{\text{seq}}(z, \{c_j\}_{j=1}^i) := E_\theta(z, \{c_j\}_{j=1}^i) + \mu \tilde{d}(z, z_{i-1}) + \gamma \sum\nolimits_{j>i} d\left(f_j(g(z); \theta), f_j(g(z_{i-1}); \theta)\right)$$

where the joint energy function $E_\theta(z, \{c_j\}_{j=1}^i)$ is from Eq. (5), and $f_j(\cdot; \theta)$ is the classifier output for the $j$-th attribute, $\tilde{d}(z, z_{i-1}) = \|g(z) - g(z_{i-1})\|_2^2 + \|z - z_{i-1}\|_2^2$ prevents $z$ from moving too far from the previous $z_{i-1}$, and the last term penalizes $z$ for changing other attributes, with $d(\cdot, \cdot)$ defined as the squared L2 norm. Note that in the $i$-th edit, we use $z_{i-1}$ as the new initial point of the sampling for a faster convergence. By default, we set $\mu = 0.04$ and $\gamma = 0.01$.

We compare our method against StyleFlow [1], the state-of-the-art in sequential editing, with two additional metrics that quantify the disentanglement of the edits: (i) the *face identity loss* (*ID*) [1, 42], which calculates the distance between the image embeddings before and after editing to measure the identity preservation. (ii) the *disentangled edit strength* (*DES*), defined as DES $= \frac{1}{n}\sum_{i=1}^n \text{DES}_i$ and $\text{DES}_i = \mathbb{E}_{p_\theta(x)}[\Delta_i - \max_{j\neq i} |\Delta_j|]$, where $\Delta_i = \frac{\text{ACC}_i - \text{ACC}_{0i}}{1 - \text{ACC}_{0i}}$ denotes the *normalized* ACC improvement, with $\text{ACC}_i$ and $\text{ACC}_{0i}$ being the ACC score of $i$-th attribute after and before the $i$-th edit, respectively. Intuitively, the maximum DES is achieved when each edit precisely control the considered attribute only but leave other attributes unchanged, leading to good disentanglement.

Table 3 shows the quantitative results, where the truncation coefficient $\psi = 0.5$, and we apply the subset selection strategy as proposed in StyleFlow [1] to alleviate the background change and the reweighting of energy functions (see Appendix A.3 for details) for better disentanglement. Our method outperforms StyleFlow in terms of disentanglement (DES), identity preservation (ID), image quality (FID) and controllability (ACC). This is confirmed by the qualitative results in Figure 3, where

Table 3: Comparison between our method and StyleFlow [1] for sequential editing on FFHQ, where we edit each attribute in the sequence of [yaw, smile, age, glasses]. Note that here we only show the final performance *after all edits*, and the results of every individual edit and ablation studies are deferred to Appendix A.8.

| Methods | All (yaw_smile_age_glasses) | | | | | | |
|---|---|---|---|---|---|---|---|
| | DES↑ | ID↓ | FID↓ | $ACC_y$↑ | $ACC_s$↑ | $ACC_a$↑ | $ACC_g$↑ |
| StyleFlow [1] | 0.569 | 0.549 | 44.13 | **0.947** | 0.773 | 0.817 | 0.876 |
| LACE-ODE | **0.735** | **0.501** | **27.94** | 0.938 | **0.956** | **0.881** | **0.997** |

Figure 3: Sequentially editing images with our method (LACE-ODE) and StyleFlow [1] on FFHQ with a sequence of [yaw, smile, age, glasses]. Our method can successfully perform each edit while less affecting the other attributes. On the contrary, StyleFlow may unintentionally modify/lose glasses, largely change the face identity, or have a smiling face when it is not supposed to.

StyleFlow usually has the following issues: (i) *changing unedited attributes*, such as accidentally modifying or losing glasses and changing face identities, and (ii) *having incomplete edits*, for example, the smiling face still appears after setting smile=0 (last row in Figure 3). On the contrary, our method suffers less from the above problems. Moreover, as the sequential editing in our method is defined by simply composing energy functions, it can incorporate novel attributes in a plug-and-play way.

## 3.3 Compositional generation

**Zero-shot generation** The goal is to generate novel images conditioned on *unseen combinations* of attributes that are not present in the training data, which is used for evaluating a model's compositionality in controllable generation. We compare our method against StyleFlow on zero-shot generation in Figure 4, where we condition on (a) {beard=1, smile=0, glasses=1, age=15} and (b) {gender=female, smile=0, glasses=1, age=10}, respectively. We can see our method still performs well in zero-shot generation while StyleFlow suffers from a severe deterioration of image quality and diversity, and almost completely fails in the controllability. Quantitatively, the ACC scores of our method are much larger than StyleFlow, as we have 0.935 vs. 0.679 (smile), and 0.982 vs. 0.694 (glasses) in the setting (a), and 0.906 vs. 0.408 (smile), and 0.939 vs. 0.536 (glasses) in the setting (b). These results clearly demonstrate the strong compositionality of our method.

**Compositions of energy functions** EBMs have shown great potential in concept compositionality with various ways of composing energy functions [12]. Our method can inherit their compositionality, while achieving high image quality. Inspired by [12], we also consider composing energy functions with three logical operators: conjunction (AND), disjunction (OR) and negation (NOT). In particular,

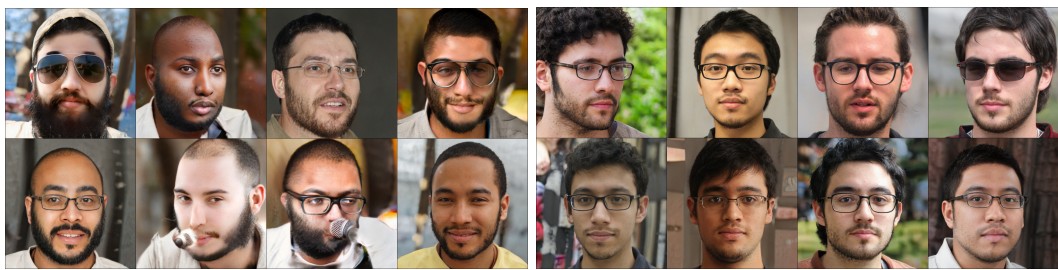

(a) `beard=1,smile=0,glasses=1,age=15` (*Left*: StyleFlow, *Right*: Ours)

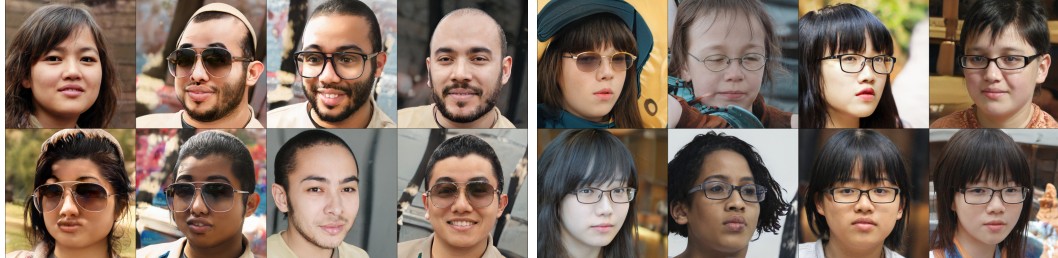

(b) `gender=female,smile=0,glasses=1,age=10` (*Left*: StyleFlow, *Right*: Ours)

Figure 4: Zero-shot conditional sampling results of our method and StyleFlow [1] on FFHQ, where none of the above two attribute combinations is seen in the training set. Our method performs well on zero-shot generation while StyleFlow almost fails by either generating low-quality images or missing the conditioning information.

given two attributes $\{c_1, c_2\}$, we can define the joint energy function of each logical operator as

$$E(z, \{c_1 \text{ AND } c_2\}) := E(c_1|g(z)) + E(c_2|g(z)) + \frac{1}{2}\|z\|_2^2$$

$$E(z, \{c_1 \text{ OR } c_2\}) := -\log\left(e^{\beta - E(c_1|g(z))} + e^{-E(c_2|g(z))}\right) + \frac{1}{2}\|z\|_2^2$$

$$E(z, \{c_1 \text{ AND } (\text{NOT } c_2)\}) := E(c_1|g(z)) - \alpha E(c_2|g(z)) + \frac{1}{2}\|z\|_2^2$$

where the AND operator actually boils down to the conditional sampling with multiple attributes, and $\alpha, \beta > 0$ are tunable hyperparameters to balance the importance of different energy functions. For more complex compositionality, we can recursively apply these logical operators.

Figure 6 demonstrates the concept compositionality of our method for `glasses` and `yaw` where we set $\beta = \ln 20$, and $\alpha = \min(\frac{0.1}{|E(z,c_2)|}, 1)$. We can see the generated images not only precisely follow the rule of the given logical operators, but are also sufficiently diverse to cover all possible logical cases. For instance, given {`glasses=1 OR yaw=front`}, some images have glasses regardless of the yaw while other images satisfy `yaw=front` regardless of the glasses. Our method also works well for the recursive combinations of logical operators, as shown in the bottom-right of Figure 6. To the best of our knowledge, our method is the first to show such strong compositionality when controllably generating photo-realistic images of resolution $1024 \times 1024$.

### 3.4 Ablation study on the ODE and LD sampler

Here, we carefully examine the ODE and LD samplers with a grid search in a large range of hyperparameter settings (see Appendix A.11 for details). Figure 5 shows the ACC and FID trade-off of the two samplers on CIFAR-10, where each dot in the figure refers to the result of a particular hyperparameter setting. After grid research, there are 81 and 104 hyperparameter settings for the ODE and LD sampler, respectively.

We can see that (i) the best ACC-FID scores (on the top-left of Figure 5) of the two samplers heavily overlap, implying that they perform equally well in their best tuned hyperparameter settings. (ii) With the concentrated ACC-FID scores for the ODE sampler on the top-left, we can see that the ODE sampler is more stable and less sensitive to the choice of hyperparameters than the LD sampler. (iii) When focusing on the top-left with ACC $\geq 95\%$ and FID $\leq 10$, the ODE sampler needs much smaller (less than

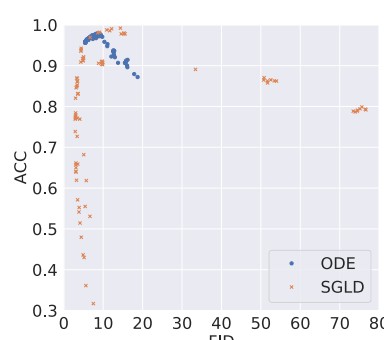

Figure 5: The ACC-FID distributions for the ODE and LD sampler, where each dot denotes each hyperparameter setting.

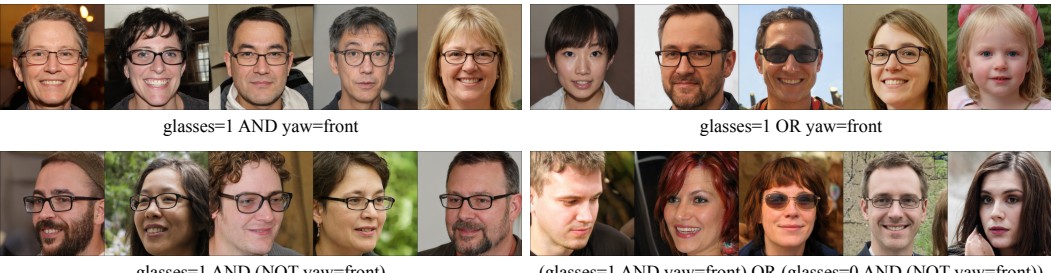

glasses=1 AND yaw=front        glasses=1 OR yaw=front

glasses=1 AND (NOT yaw=front)     (glasses=1 AND yaw=front) OR (glasses=0 AND (NOT yaw=front))

Figure 6: Compositions of energy functions in our method with different logical operators: conjunction (AND), disjunction (OR), negation (NOT), and their recursive combinations on FFHQ of resolution 1024×1024.

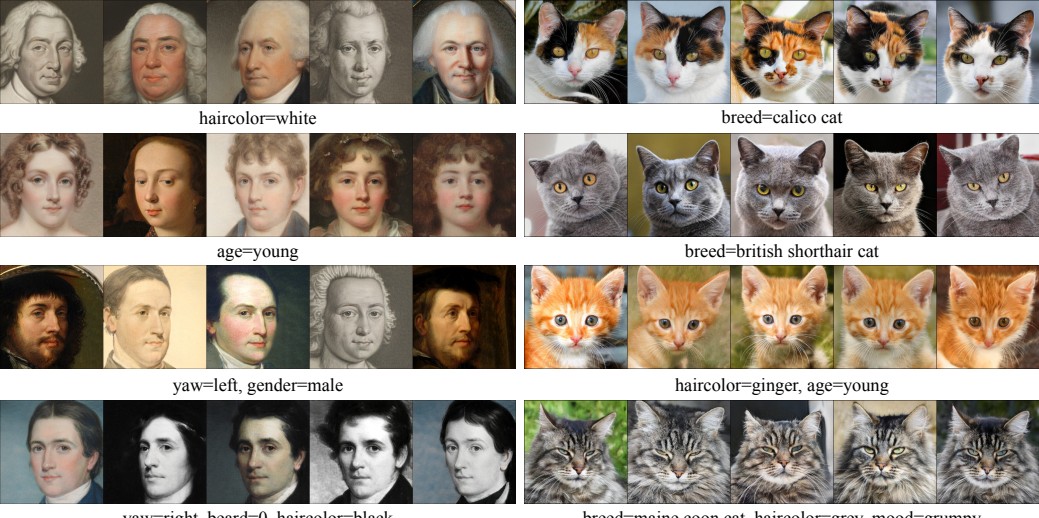

haircolor=white          breed=calico cat

age=young          breed=british shorthair cat

yaw=left, gender=male       haircolor=ginger, age=young

yaw=right, beard=0, haircolor=black    breed=maine coon cat, haircolor=grey, mood=grumpy

(a) MetFaces of resolution 1024×1024      (b) AFHQ-Cats of resolution 512×512

Figure 7: Conditional sampling of our method (LACE-ODE) on (a) MetFaces of resolution 1024×1024 and (b) AFHQ-Cats of resolution 512×512, respectively. The generated images are conditioned on either a single attribute or a combination of multiple attributes.

21%) number of minimum average steps than the LD sampler (41.8 vs. 200). It shows that given the similar performance, the ODE sampler tends to be more efficient due to its adaptivity in step sizes.

### 3.5 Results on other high-resolution images

**MetFaces [27]** By applying the image classifier pre-trained on the FFHQ data to label the painting faces data (i.e., MetFaces), we train a latent classifier in the $w$-space of StyleGAN-ADA on MetFaces. Accordingly, our method can effectively control the generation of painting faces with one or multiple attributes, as shown in Figure 7(a). It demonstrates the robustness of our method to the classification noise and its generalization ability regarding the domain gap between photos and paintings.

**AFHQ-Cats [11]** Since the original AFHQ-Cats dataset does not contain the ground-truth attributes (i.e., breed, haircolor, and age, etc.), we have to rely on extra annotators for efficient labeling. Inspired by [15], we apply the CLIP [43] to annotate the AFHQ-Cats data by designing proper prompts that contain the controlling attributes. Similarly, we then train a latent classifier in the $w$-space of StyleGAN-ADA on AFHQ-Cats to apply our method for controllable generation. As shown in Figure 7(b), we can effectively control the generation of cat images based on the CLIP annotations.

## 4 Related work

**Conditional generative models** Significant efforts for conditional sampling and image editing have been invested in conditional generative models. In these models, various conditional information can be used to guide the generation process, such as class labels [36, 32], attribute codes [9, 38], source images [10], text descriptions [44], and semantic maps [41]. Training these models is costly, in particular for generating high-resolution images. Additionally, adding or modifying the conditional information requires re-training the whole network. On the contrary, our method is based on pre-trained unconditional models, and it requires little effort to introduce new conditional information.

**Latent code manipulation for image editing**   An alternative approach to conditional GANs is to manipulate the latent code of pre-trained unconditional models, in particular GANs. Some works explore linear manipulations of latent code [26, 20, 46], while others consider more complicated nonlinear manipulations [37, 16, 1, 42]. Similar to our work, these methods can discover the directions that often correspond to meaningful semantic edits, and need not train conditional generative models from scratch. However, they tend to have issue with compositionality. For instance, StyleFlow [1] specifies a fixed length of attributes during training, limiting its ability to generalize to new attributes and novel combinations (Section 3.3). Notably, [37, 16] share the similar idea of optimizing latent variables of generators with a classifier. But with the intuition of using generative models as a powerful prior, their classifiers are defined in the pixel space only, making their sampling more challenging. Instead, our method is formulated in a principled way from the EBM perspective (that unifies these methods) and can be defined in various spaces (pixel / $w$ / $z$) of a generator.

**EBMs and score-based models**   EBMs [34] and score-based models [48] have been widely used for image generation [59, 12, 49, 51], due to their flexibility in probabilistic generative modeling. Recently, many works have applied them for controllable and compositional generation. [18] proposes the joint EBM and shows its ability in class-conditional generation. [12] explores the compositionality of conditional EBMs in controllable generation. [51] proposes a unified framework of score-based models from the SDE perspective, and performs controllable generation by using both a conditional reverse-time SDE and its corresponding ODE. Unlike our method that works in the latent space, they all formulate energy functions or score functions in the pixel space, making them both more challenging to train and orders of magnitude slower to sample for high-resolution images.

Also, many other works have built EBMs or score-based models in the latent space of generative models, most of which, however, focus on improving generation quality with EBMs as a structural prior [2, 4, 19, 56, 55, 58, 53]. More similar to our work, [60, 40] considers using the conditional EBMs as a latent prior (conditioning on labels or attributes) of a generator for controllable generation. A key difference is that they require inferring latent variables from data or sampling from the EBMs using LD during training while we do not need to. This advantage distinguishes our method from these prior methods regarding training speed and inference efficiency.

## 5   Discussions and limitations

We proposed a novel formulation of a joint EBM in the latent space of pre-trained generative models for controllable generation. Based on our formulation, all we need for controllability is to train an attribute classifier, and sampling is done in the latent space. Moreover, we proposed a more stable and adaptive sampling method by formulating it as solving an ODE. Our experimental results showed that our method is fast to train and efficient to sample, and it outperforms state-of-the-art techniques in both conditional sampling and sequential editing. With our strong performance in the zero-shot generation with unseen attribute combinations and compositions of energy functions with logical operators, our method also demonstrated compositionality in generating high-quality images.

One limitation of this work is that the generation quality is limited by the generative power of the underlying pre-trained generator. Since our EBM can be applied to any latent-variable model including GANs and VAEs, it is interesting to apply our method to different generative models based on downstream tasks. Another limitation is that training the attribute classifier requires the availability of attribute labels. We believe that advanced semi-supervised and self-supervised techniques [47, 8] can improve the classifier training while reducing the dependency on labels.

## 6   Broader impact

The method presented in this work enables high-quality controllable image synthesis, built upon pre-trained generative models (e.g., StyleGAN2 [29]). Technically, it inherits the compositionality of EBMs and overcomes their difficulty in generating high-resolution images. Since our method does not train any conditional generator from scratch, it significantly reduces the computational cost of training generators for new conditioning attributes.

On the application side, it shares with other image synthesis tools similar potential benefits and risks, which have been discussed extensively in [5, 52]. Our method does not produce new images but only guides the generation process of existing generators. Thus, it inherits potential biases from the pre-trained generators, e.g. StyleGAN2 trained on the FFHQ dataset [28]. On the other hand, by providing a semantic control of image generation with strong compositionality, our method can be used to discover and proactively reduce unknown biases in existing generators.

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
