# A Appendix

## A.1 Derivation of the joint distribution in latent space

Our proof closely follows [6] from the rejection sampling perspective. Before the derivation, we introduce the following lemma about the resulting probability distribution in rejection sampling:

**Lemma 1** *(Lemma 1 in [6]) Given a probability distribution $p(x)$ where $x \in \mathcal{X}$ and a measurable function $r : \mathcal{X} \to [0, 1]$, the rejection sampling with the proposal distribution $p(x)$ and acceptance probability $r(x)$ generates $x$ samples following from the distribution $q(x)$, which satisfies $q(x) = p(x)r(x)/Z_0$ where $Z_0 = \mathbb{E}_{x \sim p(x)}[r(x)]$.*

*Proof:* See the proof of Lemma 1 in [6]. □

Recall that we define the generative model as

$$p_\theta(x, c) = p_g(x)e^{-E_\theta(c|x)}/Z \tag{8}$$

where $p_g(x)$ is an implicit distribution defined by a pre-trained generator $g$ in the form of $x = g(z)$ with the latent variable $z \in \mathcal{Z}$ and the data $x \in \mathcal{X}$, and $Z$ is a normalization constant.

When conditioned on the attribute $c$, we have $p_\theta(x|c) = \frac{p_\theta(x,c)}{p_\theta(c)}$. Similar to [6], if we do rejection sampling with the proposal distribution $p_g(x)$ and the acceptance probability $\frac{p_\theta(x|c)}{M(c)p_g(x)}$, where $M(c)$ is a constant (w.r.t. $x$) satisfying $M(c) \geq \frac{p_\theta(x|c)}{p_g(x)}$, we get samples $x$ from $p_\theta(x|c)$. As $p_g(x)$ is an implicit distribution induced by the generator $x = g(z)$, this rejection sampling on $p_g(x)$ is equivalent to a rejection sampling on the prior $p(z)$ in the latent space and then applying $x = g(z)$.

Specifically, the corresponding rejection sampling in the latent space has the proposal distribution $p(z)$ and the acceptance probability

$$r_\theta(z, c) = \frac{p_\theta(g(z)|c)}{M(c)p_g(g(z))} = \frac{e^{-E_\theta(c|g(z))}}{M(c)p_\theta(c)Z} \tag{9}$$

where the second equation follows from Eq. (8). From Lemma 1, we have that conditioned on the attribute $c \sim p_\theta(c)$, the resulting probability $p_\theta(z|c)$ of the accepted $z$ samples in the above rejection sampling procedure satisfies

$$p_\theta(z|c) = p(z)r_\theta(z, c)/Z'(c) \tag{10}$$

where $Z'(c) := \mathbb{E}_{z \sim p(z)}[r_\theta(z, c)]$. By substituting Eq. (9) into Eq. (10), we have

$$p_\theta(z|c) = \frac{p(z)e^{-E_\theta(c|g(z))}}{p_\theta(c)Z''(c)} \tag{11}$$

where $Z''(c) := M(c)ZZ'(c)$, and from Eq. (9) we further have

$$
\begin{aligned}
Z''(c) &= \mathbb{E}_{z \sim p(z)}[e^{-E_\theta(c|g(z))}]/p_\theta(c) \\
&= \mathbb{E}_{x \sim p_g(x)}[e^{-E_\theta(c|x)}]/p_\theta(c) \\
&= \frac{Z}{p_\theta(c)} \int p_\theta(x, c)dx \\
&= Z
\end{aligned}
\tag{12}
$$

where the second equation follows from the change of variables with $x = g(z)$, and the third equation follows from marginalizing $x$ in Eq. (8). Thus, Eq. (11) yields

$$p_\theta(z, c) = p(z)e^{-E_\theta(c|g(z))}/Z \tag{13}$$

which concludes the derivation. □

## A.2 Derivation of Eq. (7)

Song et al. [51] define a forward diffusion process that maps samples $x_0 \sim p_{\text{data}}$ to $x_T \sim p_T = N(0, I)$ using the VP-SDE:

$$dx = -\frac{1}{2}\beta(t)xdt + \sqrt{\beta(t)}dw \tag{14}$$

for $t \in [0, T]$. They also show that a generative SDE can be defined by:

$$dx = -\frac{1}{2}\beta(t)[x + 2\nabla_x \log p_t(x)]dt + \sqrt{\beta(t)}d\bar{w} \tag{15}$$

with time flowing backward from $T$ to 0 and reverse standard Wiener process $\bar{w}$. For conditional generation, the generative SDE above becomes:

$$dx = -\frac{1}{2}\beta(t)[x + 2\nabla_x \log p_t(x,c)]dt + \sqrt{\beta(t)}d\bar{w} \tag{16}$$

where $p_t(x, c)$ is the join distribution of data and attribute at time $t$.

Song et al. [51] show that there exists an equivalent ODE whose trajectories share the same joint probability densities $p_t(x(t), c)$ with the reverse SDE defined in Eq. (16):

$$dx = -\frac{1}{2}\beta(t)\left[x + \nabla_x \log p_t(x,c)\right]dt \tag{17}$$

where the idea is to use the Fokker-Planck equation [39] to transform an SDE to an ODE (see Appendix D.1 for more details in [51]).

Song et al. [51] define $\nabla_x \log p_t(x(t), c) := \nabla_x \log p_t(x(t)) + \nabla_x \log p_t(c|x(t))$ which requires (i) estimating the score function $\nabla_x \log p_t(x(t))$, and (ii) training a *time-variant* classifier $p_t(c|x(t))$ in the $x$-space for $\nabla_x \log p_t(c|x(t))$, both of which make the training and inference challenging. Instead, in our framework, we solve the ODE in the $z$-space and transfer the $z$ samples to the data space with the generator $g$. Specifically with $p_t(z(t), c) \propto e^{-E_t(c|g(z(t)))+\log p_t(z(t))}$, we have

$$dz = -\frac{1}{2}\beta(t)\left[z + \nabla_z \log p_t(z) - \nabla_z E_t(c|g(z))\right]dt \tag{18}$$

Since the distribution of latent variable $z$ in most generative models satisfies $p_0(z(0)) = \mathcal{N}(0, I)$, diffusing it with VP-SDE will not change its distribution at time $t$, i.e., $p_t(z(t)) = \mathcal{N}(0, I)$ [51]. Since $p_t(z(t))$ is time-invariant and the generator $g$ is fixed, the classifier $p_t(c|g(z(t)))$ receives input $g(z(t))$ with a time-invariant distribution. Thus, we assume that the classifier is also time-invariant, and so is the energy function $E_t(c|g(z(t))) := E(c|g(z(t)))$. Therefore, the above ODE becomes:

$$dz = \frac{1}{2}\beta(t)\nabla_z E(c|g(z))dt = \frac{1}{2}\beta(t)\sum_{i=1}^n \nabla_z E(c_i|g(z))dt \tag{19}$$

which concludes the derivation. $\square$

### A.3 More details of experimental setting

**Training hyperparameters** For the classifier $f_i(x; \theta)$, we use a four-layer MLP as the network architecture when it is trained in the $z$-space or the $w$-space of StyleGAN2, as shown in Table 4. We train the classifier with the Adam optimizer [30] for 100 epochs using a staircase decay schedule. When we consider the classifier in the pixel space, we directly use the pre-trained WideResNet-28-10 [61] with no batch normalization as the network architecture.

Table 4: The four-layer MLP architecture as the attribute classifier when it is trained in the $z$-space or the $w$-space of StyleGAN2. Its output dimension depends on the number of class predictions for the targeted attributes.

| input: $z \in \mathbb{R}^{512}$ or $w \in \mathbb{R}^{512}$ |
| --- |
| Linear 384, LeakyReLU |
| Linear 256, LeakyReLU |
| Linear 128, LeakyReLU |
| Linear #logits |

**Inference hyperparameters** By default, we use the 'dopri5' ODE solver with the absolute and relative tolerances (`atol`, `rtol`) being set to (1e-3, 1e-3) in LACE-ODE. The time-variant diffusion coefficient $\beta(t)$ in the ODE sampler has the form that $\beta(t) = \beta_{\min} + (\beta_{\max} - \beta_{\min})t$, where $\beta_{\min} = 0.1$ and $\beta_{\max} = 20$, and $t \in [0, 1]$. Similar to the prior work [18], for the LD sampling in LACE-LD, the step size $\eta$ and the standard deviation of $\epsilon_t$ are chosen separately for faster training, although it results in a biased sampler [18, 14]. By default, we set the number of steps to be 100, step size $\eta = 0.01$ and standard deviation of $\epsilon_t$ to be 0.01 in LACE-LD.

We use StyleGAN-ADA as the pre-trained generator for experiments on CIFAR-10 and StyleGAN2 for experiments on FFHQ. StyleGAN-ADA shares the same network architecture with StyleGAN2, where the truncation trick can be applied in the $w$-space for better image quality [28]. Since our EBM formulation does not modify the generator architecture, we can also apply the truncation in the $w$-space during sampling. By default, we use the truncation coefficient $\psi = 0.7$ in our method.

**Hardware**    We ran all experiments on one single NVIDIA V100 GPU with 32GB memory size.

**Data preparation**    To train attribute classifier in the $z$-space or the $w$-space of StyleGAN2, we first need a training set of the pairs $(z, c)$ or $(w, c)$, where $c$ represents the class label in CIFAR-10 and the attribute code in FFHQ. For the experiments on CIFAR-10, we first sample 50k $z$ latent variables from the standard Gaussian and pass them to the StyleGAN2 generator to get 50k $w$ samples (i.e., the output of the mapping network) and 50k images (i.e., the output of the synthesis network). Then, we label the 50k images using a pre-trained DenseNet [25] image classifier with an error rate 4.54% on CIFAR-10. Accordingly, we get 50k $(z, c)$ pairs and 50k $(w, c)$ pairs as the training sets.

For the experiments on FFHQ, we directly use the 10k $(w, c)$ pairs created by the StyleFlow paper [1] to train the attribute classifier. We use 12 attributes in most of our experiments, including 5 discrete attributes: `smile`, `glasses`, `gender`, `beard`, `haircolor`, and 7 continuous attributes: `yaw`, `age`, `pitch`, `bald`, `width`, `light0`, `light3`. Usually for binary discrete attributes, we denote "1" as the presence and "0" as the absence. For instance, `glasses=1` means wearing glasses and `glasses=0` means no glasses. Similarly, `smile=1` means smiling and `smile=0` means no smiling. For continuous attributes, we normalize their values to the range of $[0, 1]$.

**Metrics**    To quantify the model performance in controllable generation, we mainly use the following two metrics: (i) *conditional accuracy* (*ACC*) to measure the controllability, and (ii) *FID* to measure the generation quality and diversity [21]. Specifically, the ACC score is calculated as follows: We first generate $N_a$ images based on randomly sampled attributes codes, and then pass these generated images to a pre-trained image classifier to predict the attribute codes. Accordingly, the ACC score reflects how accurately the predicted attribute codes match the sampled ground-truth ones.

For the experiments on CIFAR-10 and FFHQ, these two metrics are evaluated in slightly different ways. First, for the ACC score on CIFAR-10, we use the aforementioned DenseNet pre-trained on CIFAR-10 as the image classifier. For each class, we uniformly sample 1k images, meaning the total generated images $N_a = 10k$. The final ACC score is then the averaged accuracy of the predicted class labels over 10 classes. For the FID score on CIFAR-10, we uniformly sample 5k images in each class and then use the total 50k images to calculate the FID.

Second, for the ACC score on FFHQ, we use the MobileNet [24] as the network backbone of the image classifier due to its small size and effectiveness in the recognition of face attributes. To improve the generalization ability of the image classifier, we first train the MobileNet with the Adam optimizer for 10 epochs on the CelebA dataset [35], and then fine-tune it on the 10k generated FFHQ image and attribute pairs for another 50 epochs. Similarly, we *uniformly* sample the attribute codes from the set of all possible combinations, and generate 1k images to compute the ACC. The final ACC score is then the averaged accuracy of the predicted attribute codes over all the sampled attribute combinations. Note that for the continuous attributes $c_i \in \mathbb{R}$, such as `yaw` and `age`, we normalize their values to the range of $[0, 1]$, and the ACC for each continuous attribute is represented by $1 - |\hat{c}_i - c_i|$ instead, where $\hat{c}_i$ is the predicted continuous attribute from the MobileNet image classifier.

For the FID score on FFHQ, we follow from StyleFlow [1] that uses 1k generated samples StyleGAN2 to compute the FID. Note that the resulting FID scores are not comparable to those reported in the original StyleGAN2 paper, as it uses 50k real FFHQ images to evaluate the FID. For the reference, the original unconditional StyleGAN2 with our evaluation protocol has FID=20.87±0.11. Besides, different from CIFAR-10 where each class has equally distributed samples, the attribute distribution in the FFHQ data is heavily imbalanced. For instance, the number of images with `glasses=0` is at least 5× larger than that with `glasses=1`. The number of images with `smile=1` is at least 3× larger than that with `smile=0`. Thus, if we uniformly sample attributes as before, the resulting generated image distribution will largely deviate from the reference data distribution, making the FID score incorrectly reflect the generation quality. To remedy this, we randomly sample attribute codes *from the training set* instead to generate 1k images for the FID evaluation on FFHQ.

### A.4    More details of baselines

We use different baselines for comparing with our method in controllable generation. The first set of baselines is the EBMs in the pixel space:

**JEM [18]**    It proposes the joint EBM framework of modelling the data and labels in the pixel space. Its training is composed of two parts: $p_\theta(c|x)$ for the classifier training and $p_\theta(x)$ for the generative

modelling. In both training and inference of JEM, the LD sampling is applied to draw samples from $p_\theta(x)$. We use the default hyperparameter settings in [18] to report its results.

**Cond-EBM [12]**    Based on conditional EBMs, it proposes different ways of composing the energy functions with logical operators for compositional generation. To train conditional EBMs, it also applies the LD sampling to draw samples from $p_\theta(x|c)$. We use the default hyperparameter settings in [12] to report its results. Particularly during inference, to improve the generation quality, we apply the following tricks [12]: (i) we combine two training checkpoints, and (ii) we run 50 LD steps followed by the data augmentations to get a good initialization of the LD sampling.

The second set of baselines is the score-based models with SDEs [51]:

**VP-SDE [51]**    In *Variance Preserving (VP) SDE* [51], the forward SDE is defined as

$$dx = -\frac{1}{2}\beta(t)xdt + \sqrt{\beta(t)}dw \tag{20}$$

where $\beta(t)$ represents a scalar time-variant diffusion coefficient and $w$ is a standard Wiener process. Then, the conditional sampling from $p_0(x|c)$ is equivalent to solving the following reverse SDE:

$$dx = -\frac{1}{2}\beta(t)[x + \nabla_x \log p_t(x,c)]dt + \sqrt{\beta(t)}d\bar{w} \tag{21}$$

where $\bar{w}$ is a standard Wiener process when time flows backwards from $T$ to 0. To sample from Eq. (21), the *Predictor-Corrector (PC)* sampler is proposed in [51]. At each time step, the numerical SDE solver first gives an estimate of the sample at the next time step, playing the role of a "predictor". Then, the score-based MCMC approach corrects the marginal distribution of the estimated sample, playing the role of a "corrector" [51].

**VE-SDE [51]**    In *Variance Exploding (VE)* SDE, the forward SDE is defined as

$$dx = \sqrt{\frac{d[\sigma^2(t)]}{dt}}dw \tag{22}$$

where $\sigma(t)$ represents a sequence of positive noise scales and $w$ is a standard Wiener process. Then, the conditional sampling from $p_0(x|c)$ is equivalent to solving the following reverse SDE:

$$dx = -\nabla_x \log p_t(x,c)d[\sigma^2(t)] + \sqrt{\frac{d[\sigma^2(t)]}{dt}}d\bar{w} \tag{23}$$

where $\bar{w}$ is a standard Wiener process when time flows backwards from $T$ to 0. According to [51], the PC sampler can also be applied to sample from Eq. (23).

To report the controllable generation results of VP-SDE and VE-SDE, we use the pre-trained models released by the official implementation (`https://github.com/yang-song/score_sde`), and also used the default sampling hyperparameters of the PC sampler: the number of predictor steps $N = 1000$, the number of corrector steps $M = 1$, and the signal-to-noise ratio $r = 0.16$.

The last set of baselines is the methods modelled in the latent space of the pre-trained generator:

**StyleFlow [1]**    It applies the conditional continuous normalizing flows (CNFs) to build an invertible mapping between the $z$-space and the $w$-space of StyleGAN2 conditioned on the attribute codes. The goal is to enable adaptive latent space vector manipulation by casting the conditional sampling problem in terms of conditional CNFs using the attributes for conditioning [1].

The conditional sampling task is straightforward: it sets the attribute code to a desired set of values, and then samples multiple $z$ variables, which are passed to the conditional CNF and the synthesis network of StyleGAN2 to get the final images. The sequential editing task is mainly composed by a sequence of Conditional Forward Editing (CFE) and Joint Reverse Encoding (JRE). Meanwhile, several hand-crafted tricks are applied to improve the editing quality, including the Edit Specific Subset Selection and re-projection of edited $w$ to the $z$-space. See the original paper [1] for details.

To get the reported results, we use the pre-trained models released by the official implementations (`https://github.com/RameenAbdal/StyleFlow`). As we keep all the hand-crafted tricks mentioned above, it implies that we actually use the StyleFlow (V2) [1] for comparison. Besides that, we use the default sampling hyperparameters. In particular, we use the adjoint method to compute the gradients and solve the ODE using 'dopri5' ODE solver, where the tolerances are set to 1e-5.

**Latent-JEM** This is a baseline we propose by modelling JEM [18] in the latent space of a pre-trained generator. Similarly, the Latent-JEM is modelled in the $w$-space of StyleGAN2. Given the joint distribution of $w$ varaible and attribute code $c$:

$$p_\theta(w, c) \propto e^{-E_\theta(w,c)}, \tag{24}$$

then we assume $E_\theta(w, c) = \sum_{i=1}^n E_\theta(w, c_i)$ (i.e., the conditional independence assumption) where

$$E_\theta(w, c_i) = \begin{cases} -f_i(x, \theta)[c_i] & \text{if } c_i \text{ is discrete} \\ \frac{1}{2\sigma^2}(c_i - f_i(x, \theta))^2 & \text{if } c_i \text{ is continuous} \end{cases} \tag{25}$$

Similarly, $f_i(x; \theta)$ is the output of a multi-class classifier mapping from $\mathcal{X}$ to $\mathbb{R}^{m_i}$ if the $i$-th attribute is discrete or a regression network mapping from $\mathcal{X}$ to $\mathbb{R}$ if it is continuous. Note that the original JEM paper [18] has only considered the discrete case, so here we propose a more generalized framework that also works for the continuous attributes.

By marginalizing out $c$ in Eq. (24), we obtain an unnormalized density model:

$$p_\theta(w) \propto e^{-E_\theta(w)}, \tag{26}$$

where the marginal energy function is given by

$$E_\theta(w) = -\sum_{i \in \mathcal{I}_{\text{dis}}} \log \sum_{c_i} \exp(f_i(x, \theta)[c_i]) \tag{27}$$

where $\mathcal{I}_{\text{dis}}$ is the index set of all discrete attributes. Similar to JEM [18], when we compute the conditional $p_\theta(c|w)$ via $p_\theta(w, c)/p_\theta(w)$ by dividing Eq. (24) to Eq. (26), the normalizing constant cancels out, yielding the standard Softmax parameterization for the discrete attributes and the squared L2 norm parameterization for the continuous attributes.

During training, we follow from [18] to optimize $p_\theta(c|w)$ using standard cross-entropy and optimize $p_\theta(w)$ using Eq. (2) with the LD where gradients are taken with respect to the marginal energy function (27). In practice, we find a trade-off between the generation quality and controllability in Latent-JEM controlled by the step size $\eta$. Thus, after a grid search, we use both two step sizes: $\eta = 0.1$ and $\eta = 0.01$ to get the reported results, while the number of LD steps $N = 200$ and the standard deviation of noise $\sigma = 0.01$ work the best for Latent-JEM.

Besides, we use the reply buffer of size 10,000 during training and inference, as suggested by [18], to improve the results of Latent-JEM on CIFAR-10. For the experiments of Latent-JEM on FFHQ, instead of sampling $w$ from an uniform distribution as the initialization point of the LD [18], we get a better initialization of $w$ by first randomly sampling $z$ from the standard Gaussian and passing $z$ to the pre-trained mapping network of StyleGAN2. By doing so, the performance of Latent-JEM on FFHQ improves significantly.

**LACE-PC** This is another baseline we propose by replacing the ODE sampler with the Predictor-Corrector (PC) sampler from the SDE perspective [51]. We keep the EBM formulation in Eq. (5) unchanged. In experiments, we first perform a grid search on the hyperparameters of the PC sampler: the number of predictor steps $N$, the number of corrector steps $M$ and the signal-to-noise ratio $r$. Similarly, we find a trade-off between generation quality and controllability in LACE-PC, controlled by the the number of predictor steps. Thus, we use both two numbers of predictor steps: $N = 100$ and $N = 200$ to get the reported results while the number of corrector steps $M = 1$ and the signal-to-noise ratio $r = 0.05$ work the best for LACE-PC.

### A.5 Which space to train the classifier?

We use StyleGAN-ADA [27] pre-trained on CIFAR-10 [33] to investigate which space works the best to train the classifier. In particular, we compare the performances of our method in three spaces of StyleGAN-ADA: $z$-space, $w$-space and pixel space (or $i$-space). The results are shown in Table 5 for the ODE and LD sampler, respectively. We can see that in different hyperparameter settings, our method works the best in the $w$-space for both samplers. The reason why $z$-space works worse is that the classifier in the $z$-space has lower accuracy than that in the more disentangled $w$-space. The fact that we get the worst performance in the $i$-space is mainly because of its difficulty in convergence. Therefore, we focus on the $w$-space to train the classifier for our method.

Table 5: The *FID* and ACC scores of the ODE and LD sampler in different spaces of StyleGAN-ADA on CIFAR-10, where "default" means the default hyperparameter setting for each sampler, "best_acc" and "best fid" denote the hyperparameter settings with the best ACC and the best FID, respectively, in grid research.

| Sampler | Space | default | | best_acc | | best_fid | |
|---|---|---|---|---|---|---|---|
| | | ACC↑ | FID↓ | ACC↑ | FID↓ | ACC↑ | FID↓ |
| ODE | $z$ | 0.929 | 7.34 | 0.933 | **7.94** | 0.912 | 6.66 |
| | $w$ | **0.971** | **6.69** | **0.979** | 8.52 | **0.957** | **5.40** |
| | $i$ | 0.473 | 20.18 | 0.473 | 20.18 | 0.413 | 9.98 |
| LD | $z$ | 0.924 | 10.27 | 0.990 | 23.62 | 0.549 | 2.93 |
| | $w$ | **0.935** | **4.34** | **0.992** | **14.36** | **0.769** | **2.89** |
| | $i$ | 0.394 | 10.85 | 0.468 | 74.76 | 0.134 | 3.28 |

## A.6 More results of conditional sampling on CIFAR-10

We report the results of our method and baselines on CIFAR-10 with error bars in Table 6. Note that in Table 6, the reported FID is slightly higher than that of the pre-trained StyleGAN-ADA [27] (FID: $2.92 \pm 0.05$). This is because our goal is to turn an unconditional generative model into a conditional one for better controllable generation, and the controllable sampling process changes the generated data distribution. Specifically, the original StyleGAN-ADA randomly samples the latent $z$ (by following a standard Gaussian) for image generation, while our method controllably samples the latent $z$ (to satisfy the conditional attribute specifications) with the ODE/LD sampler. The resulting data distributions of the two sampling methods will be different, thus making the FID different.

The visual samples of our method (LACE-ODE) and baselines conditioned on each class of CIFAR-10 can be seen in Figure 8 and Figure 9.

Table 6: Comparison of our method and baselines for conditional sampling on CIFAR-10. For notations, Train – training time, Infer – inference time (m: minute, s: second), which refer to the single GPU time for generating a batch of 64 images, $\eta$ is the LD step size, and $N$ is the number of predictor steps in the PC sampler.

| Methods | Train | Infer | FID↓ | ACC↑ |
|---|---|---|---|---|
| JEM [18] | 2160m | 135s | $52.35_{\pm.09}$ | $0.645_{\pm.008}$ |
| Cond-EBM [12] | 2280m | 24.5s | $41.72_{\pm.01}$ | $0.792_{\pm.003}$ |
| VP-SDE [51] | 52800m | 438s | $19.13_{\pm.04}$ | $0.643_{\pm.003}$ |
| VE-SDE [51] | 52800m | 448s | $\mathbf{2.97}_{\pm.04}$ | $0.662_{\pm.002}$ |
| Latent-JEM ($\eta$=0.1) | 21m | 0.63s | $8.75_{\pm.13}$ | $0.950_{\pm.003}$ |
| Latent-JEM ($\eta$=0.01) | 21m | 0.63s | $5.65_{\pm.09}$ | $0.821_{\pm.001}$ |
| LACE-PC ($N$=100) | **4m** | 0.84s | $\mathbf{2.99}_{\pm.01}$ | $0.747_{\pm.001}$ |
| LACE-PC ($N$=200) | **4m** | 1.86s | $\mathbf{2.94}_{\pm.02}$ | $0.722_{\pm.001}$ |
| LACE-LD | **4m** | 0.68s | $4.30_{\pm.05}$ | $0.939_{\pm.002}$ |
| LACE-ODE | **4m** | **0.50s** | $6.63_{\pm.06}$ | $\mathbf{0.972}_{\pm.001}$ |

## A.7 More results of conditional sampling on FFHQ

We report the results of our method and baselines on the `glasses` and `gender_smile_age` of FFHQ with error bars in Table 7. The 1024×1024 conditional sampling visual samples of our method (LACE-ODE) and StyleFlow conditioned on {glasses=1} and {gender=female,smile=1,age=55} of FFHQ can be seen in Figure 10 and Figure 11, respectively.

**On reducing inference time** As we can see from Table 7, the inference time of our method increases with the number of attributes. In our current setting, each attribute classifier is parametrized by a separate (384-256-128) MLP network (Table 4 in the Appendix). That is, when conditioning on $n$ attributes, we have $n$ separate MLP networks. We found that the inference time of our method largely depends on the number of MLP networks. Accordingly, if we use a single MLP network with the same size and multiple prediction heads, each of which corresponds to one attribute, we can reduce the inference time without sacrificing the controllable generation performance.

We run our method for conditional sampling with the increasing number of attributes (1-5). Without loss of generality, we consider the test case: "glasses" (1), "age, glasses" (2), "smile, age, glasses" (3), "gender, smile, age, glasses" (4), "yaw, gender, smile, age, glasses" (5). The inference time for different numbers of attributes is listed in Table 8(a). Note that "separate" denotes the current setting where we use $n$ separate MLP networks for $n$ attributes, and "single" denotes the new setting where we use a single MLP network with the same size and $n$ prediction heads for $n$ attributes. We can see that although the inference time increases with the number of attributes in both cases, the new setting ("single") has much smaller inference time,

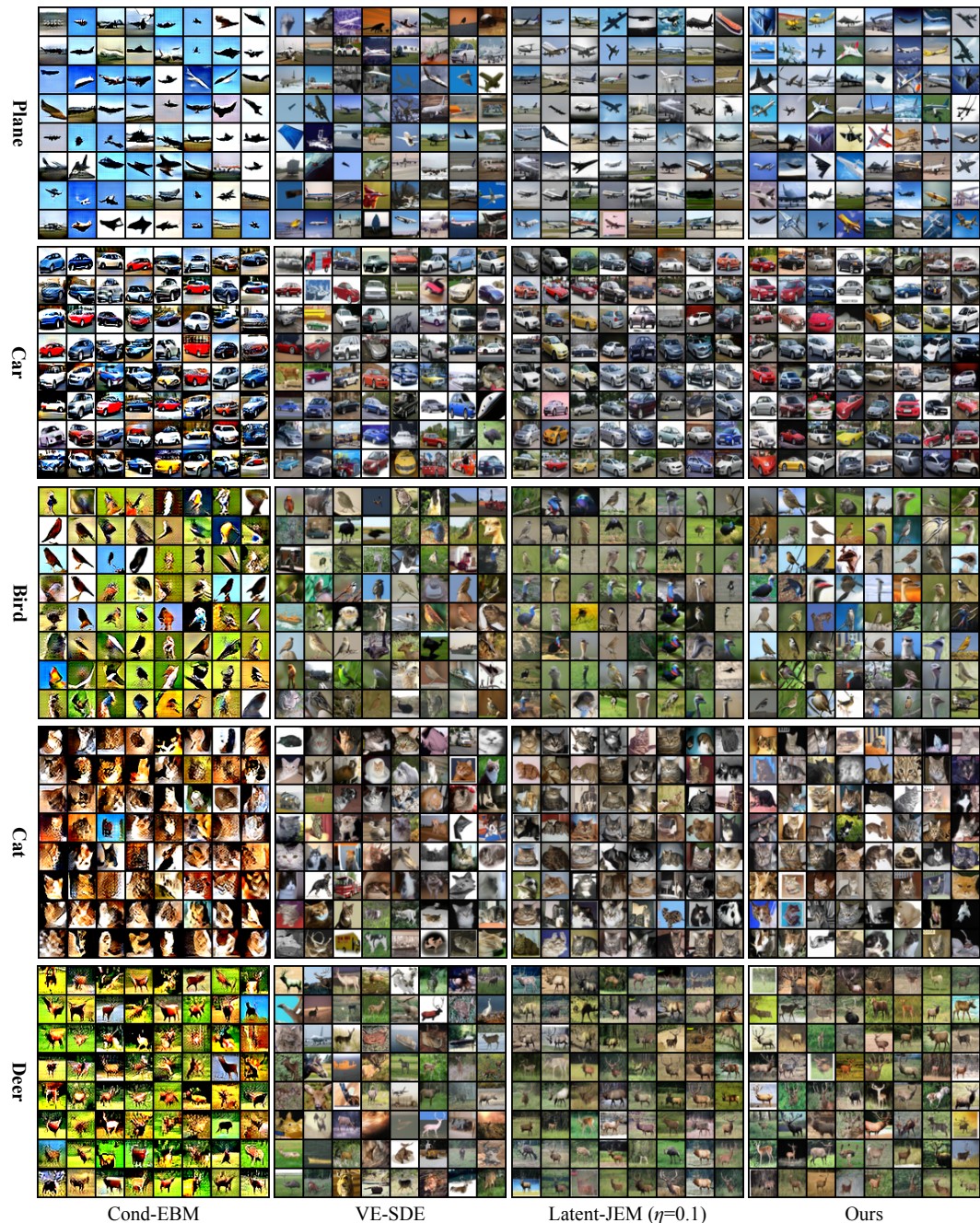

Figure 8: Conditionally generated images of our method (LACE-ODE) and baselines on each class of CIFAR-10 (0-4): plane, car, bird, cat, and deer. We can see that our method can achieve good controllability with high image quality and diversity. On the contrary, Cond-EBM suffers from the poor image quality and diversity, VE-SDE suffers from the poor controllability (with many samples inconsistent with the given class label), and the proposed baseline Latent-JEM tends to have worse image diversity than ours.

and the advantage becomes larger with more attributes. Meanwhile, the performances remain similar. For instance, in the case of conditioning "yaw, gender, smile, age, glasses" (5), the ACCs of the two settings "separate" and "single" are shown in Table 8(b).

## A.8 More results of sequential editing

In sequential editing, we apply the *subsection selection* strategy as proposed in StyleFlow [1] to alleviate the background change and the *reweighting of energy functions* to improve the disentanglement quality. We now introduce them in the following.

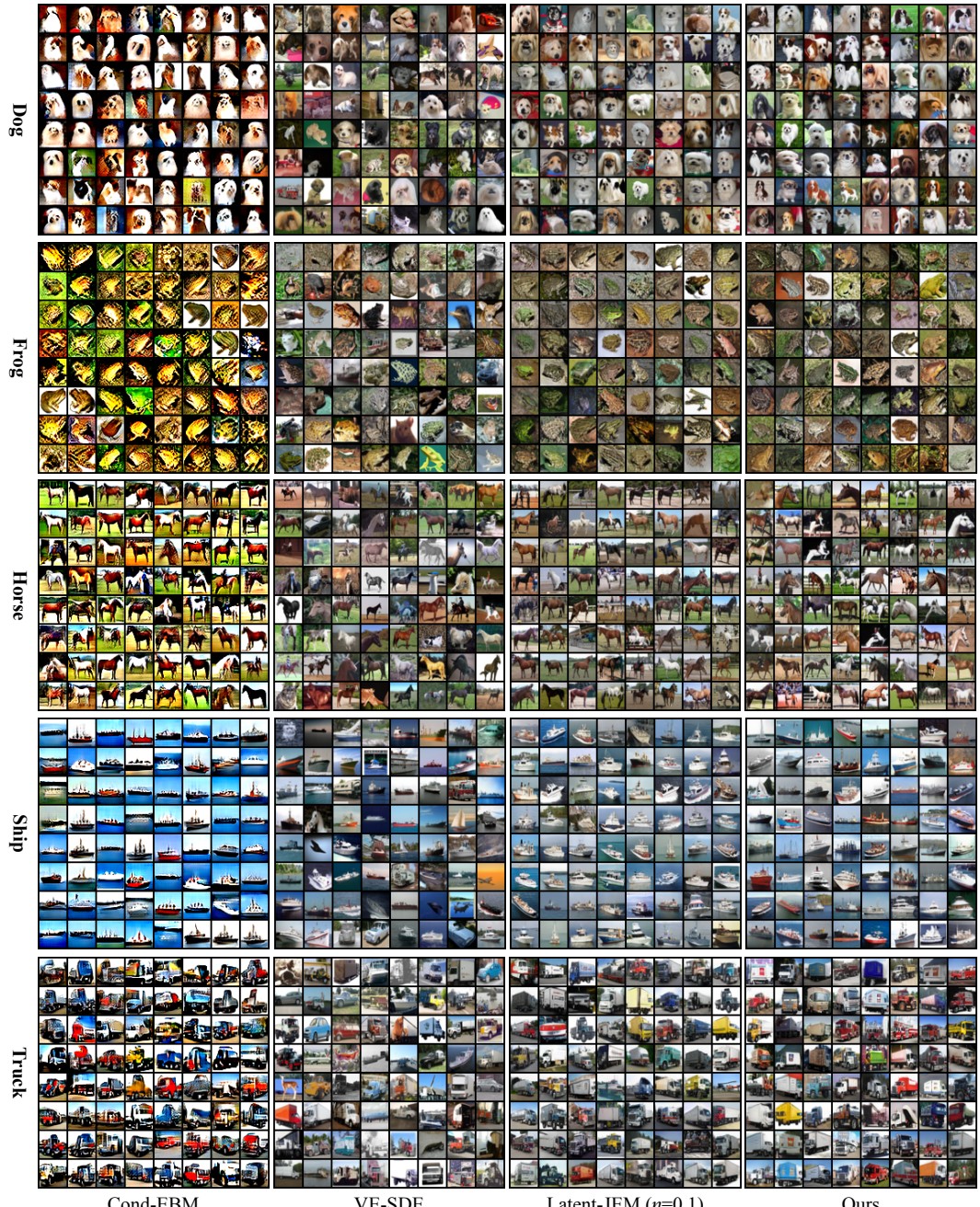

Figure 9: Conditionally generated images of our method (LACE-ODE) and baselines on each class of CIFAR-10 (5-9): dog, frog, horse, ship, and truck. We can see that our method can achieve good controllability with high image quality and diversity. On the contrary, Cond-EBM suffers from the poor image quality and diversity, VE-SDE suffers from the poor controllability (with many samples inconsistent with the given class label), and the proposed baseline Latent-JEM tends to have worse image diversity than ours.

**Subsection selection**    By observing the hierarchical structure of StyleGAN2 [29], we can apply the updated $w$ only to a subset (with different indices) of the $W+ \in \mathbb{R}^{18 \times 512}$ space, depending on the nature of each edit [1]. For instance, the head pose (such as `yaw` and `pitch`) is a coarse-grained feature and is expected to only affect the early layers of the StyleGAN2 generator. Thus, it will cause less unintentional changes (such as background and other fine-grained attributes) by applying the updated $w$ to the early layers only during editing the head pose. StyleFlow has empirically identified the index subsets of the edits that work the best for their method, including `smile` $(4 - 5)$, `yaw` $(0 -$

Table 7: Comparison between our method and baselines for conditional sampling on the `glasses` and `gender_smile_age` of FFHQ, respectively. For notations, Train – training time, Infer – inference time (m: minute, s: second), which refer to the single GPU time for generating a batch of 16 images, $\eta$ is the LD step size, and $N$ is the number of predictor steps in the PC sampler.

| Methods | Train | glasses | | | gender_smile_age | | | | |
|---|---|---|---|---|---|---|---|---|---|
| | | Infer | FID↓ | ACC$_{gl}$↑ | Infer | FID↓ | ACC$_{ge}$↑ | ACC$_s$↑ | ACC$_a$↑ |
| StyleFlow [1] | 50m | **0.61s** | 42.08$_{\pm.38}$ | 0.899$_{\pm.007}$ | **0.61s** | 43.88$_{\pm.73}$ | 0.718$_{\pm.031}$ | 0.870$_{\pm.010}$ | 0.874$_{\pm.009}$ |
| Latent-JEM ($\eta$=0.1) | 15m | 0.69s | 22.83$_{\pm.19}$ | 0.765$_{\pm.012}$ | 0.93s | 22.74$_{\pm.08}$ | 0.878$_{\pm.001}$ | 0.953$_{\pm.005}$ | 0.843$_{\pm.001}$ |
| Latent-JEM ($\eta$=0.01) | 15m | 0.69s | 21.58$_{\pm.10}$ | 0.750$_{\pm.004}$ | 0.93s | **21.98$_{\pm.14}$** | 0.755$_{\pm.003}$ | 0.946$_{\pm.009}$ | 0.831$_{\pm.002}$ |
| LACE-PC ($N$=100) | **2m** | 1.29s | 21.48$_{\pm.26}$ | 0.943$_{\pm.005}$ | 2.65s | 24.31$_{\pm.36}$ | 0.951$_{\pm.007}$ | 0.922$_{\pm.007}$ | 0.896$_{\pm.001}$ |
| LACE-PC ($N$=200) | **2m** | 2.20s | 21.38$_{\pm.37}$ | 0.925$_{\pm.003}$ | 4.62s | 23.86$_{\pm.08}$ | 0.949$_{\pm.004}$ | 0.914$_{\pm.008}$ | 0.894$_{\pm.001}$ |
| LACE-LD | **2m** | 1.15s | **20.92$_{\pm.15}$** | **0.998$_{\pm.001}$** | 2.40s | 22.97$_{\pm.14}$ | 0.955$_{\pm.004}$ | 0.960$_{\pm.002}$ | **0.913$_{\pm.001}$** |
| LACE-ODE | **2m** | 0.68s | 20.93$_{\pm.14}$ | **0.998$_{\pm.001}$** | 4.81s | 24.52$_{\pm.94}$ | **0.969$_{\pm.004}$** | **0.982$_{\pm.006}$** | **0.914$_{\pm.001}$** |

Table 8: (a) Inference time (Infer) vs. number of attributes (#attributes), and (b) ACCs of the two settings "separate" and "single" in the case of conditioning "`yaw, gender, smile, age, glasses`" (5), where "separate" means that we use $n$ separate MLP networks for $n$ attributes (i.e., the current setting), and "single" means that we use a single MLP network with the same size and $n$ prediction heads for $n$ attributes (i.e., the new setting). We can see that the new setting ("single") has much smaller inference time.

(a) Inference time vs. number of attributes

| #attributes | 1 | 2 | 3 | 4 | 5 |
|---|---|---|---|---|---|
| Infer ("separate") | 0.68 | 2.34 | 4.36 | 6.65 | 7.84 |
| Infer ("single") | 0.68 | 1.70 | 2.25 | 2.58 | 2.63 |

(b) ACCs of the two settings: "separate" and "single"

| attribute name | yaw | gender | smile | age | glasses |
|---|---|---|---|---|---|
| ACC ("separate") | 0.927 | 0.956 | 0.953 | 0.897 | 0.994 |
| ACC ("single") | 0.904 | 0.973 | 0.954 | 0.892 | 0.984 |

3), `pitch` $(0-3)$, `age` $(4-7)$, `gender` $(0-7)$, `glasses` $(0-5)$, `bald` $(0-5)$ and `beard` $(5-7$ and $10)$. To keep it simple, we directly use the above index subsets of the edits for our method.

**Reweighting of energy functions**  In reweighting of energy functions, we slightly modify the term $E_\theta(z, \{c_j\}_{j=1}^i)$ in the joint energy function of the $i$-th edit into

$$E_\theta(z, \{c_j\}_{j=1}^i) = \alpha_0 \sum_{j=1}^{i-1} E_\theta(c_j|g(z)) + \alpha_1 E_\theta(c_i|g(z)) + \frac{1}{2}\|z\|_2^2$$

where we introduce two reweighting coefficients: $\alpha_0$ for previous edited attributes and $\alpha_1$ for the current $i$-th attribute. If we set $\alpha_0 = \alpha_1 = 1$, the above equation reduces to Eq. (5). We find that slightly increasing $\alpha_1$ and decreasing $\alpha_0$ can make our method pay more attention the current edit while less modifying previously edited attributes. In experiments, we set $\alpha_0 = 0.2$, and we set $\alpha_1 = 10$ for continuous attributes and $\alpha_1 = 5$ for discrete attributes.

We report both the final results *after all edits* and the results of every individual edit with error bars in Table 9, where we edit the attributes [`yaw`, `smile`, `age`, `glasses`] in a sequential order. Besides, we also perform ablation studies on the impact of subset selection and reweighting of energy functions (see Appendix A.3 for details) on our method. From Table 9, we can see that adding subset selection or reweighting of energy functions does not change much the final ACC and FID scores. However, the disentanglement quality (DES) and identity preservation (ID) both get improved, after adding both subset selection and reweighting of energy functions.

The 1024×1024 sequential editing visual samples of our method (LACE-ODE) and StyleFlow on FFHQ with a sequence of [`yaw,smile,age,glasses`] can be seen in Figure 12.

**Randomize ordering of attributes**  We also randomly perturb the ordering of attributes and report the quantitative results in Table 10, where we edit the attributes [`age`, `yaw`, `glasses`, `smile`] in a sequential order. The results remains similar to Table 9: 1) our method largely outperform StyleFlow regarding editing quality and image quality, and 2) adding both subset selection and reweighting of energy functions can largely the disentanglement quality (DES).

## A.9   More results of compositional generation

### A.9.1   Zero-shot generation

The 1024×1024 visual samples of our method (LACE-ODE) and StyleFlow in zero-shot generation on the unseen attribute combinations {`beard=1,smile=0,glasses=1,age=15`} and

Table 9: Comparison between our method and StyleFlow [1] for sequential editing on FFHQ, where we edit each attribute of [yaw, smile, age, glasses] in a sequential order. Note that "w/o ss" means no subset selection, "w/o rw" means no reweighting of energy functions, and $\psi$ denotes the truncation coefficient of StyleGAN2.

| Methods | +yaw DES$_1\uparrow$ | ID$\downarrow$ | +smile DES$_2\uparrow$ | ID$\downarrow$ | +age DES$_3\uparrow$ | ID$\downarrow$ |
|---|---|---|---|---|---|---|
| StyleFlow | $0.568_{\pm.012}$ | $0.188_{\pm.014}$ | $0.570_{\pm.029}$ | $\mathbf{0.062}_{\pm\mathbf{.001}}$ | $0.398_{\pm.004}$ | $0.327_{\pm.015}$ |
| LACE-ODE ($\psi$=0.5, w/o ss, w/o rw) | $0.534_{\pm.016}$ | $0.179_{\pm.011}$ | $0.745_{\pm.017}$ | $0.117_{\pm.006}$ | $0.381_{\pm.010}$ | $0.175_{\pm.008}$ |
| LACE-ODE ($\psi$=0.5, w/o rw) | $0.475_{\pm.012}$ | $\mathbf{0.141}_{\pm\mathbf{.012}}$ | $0.633_{\pm.029}$ | $0.103_{\pm.006}$ | $0.260_{\pm.039}$ | $\mathbf{0.151}_{\pm\mathbf{.008}}$ |
| LACE-ODE ($\psi$=0.5) | $\mathbf{0.623}_{\pm\mathbf{.014}}$ | $0.204_{\pm.016}$ | $\mathbf{0.875}_{\pm\mathbf{.026}}$ | $0.082_{\pm.007}$ | $\mathbf{0.453}_{\pm\mathbf{.016}}$ | $0.197_{\pm.012}$ |
| LACE-ODE ($\psi$=0.7) | $0.559_{\pm.015}$ | $0.211_{\pm.011}$ | $0.825_{\pm.024}$ | $0.091_{\pm.007}$ | $0.408_{\pm.005}$ | $0.216_{\pm.011}$ |

| +glasses DES$_4\uparrow$ | ID$\downarrow$ | All DES$\uparrow$ | ID$\downarrow$ | FID$\downarrow$ | ACC$_y\uparrow$ | ACC$_s\uparrow$ | ACC$_a\uparrow$ | ACC$_g\uparrow$ |
|---|---|---|---|---|---|---|---|---|
| $0.741_{\pm.033}$ | $\mathbf{0.188}_{\pm\mathbf{.006}}$ | $0.569_{\pm.009}$ | $0.549_{\pm.016}$ | $44.13_{\pm1.62}$ | $\mathbf{0.947}_{\pm\mathbf{.004}}$ | $0.773_{\pm.022}$ | $0.817_{\pm.007}$ | $0.876_{\pm.009}$ |
| $0.956_{\pm.018}$ | $0.213_{\pm.016}$ | $0.654_{\pm.009}$ | $0.523_{\pm.005}$ | $27.46_{\pm0.16}$ | $0.941_{\pm.003}$ | $0.968_{\pm.017}$ | $\mathbf{0.897}_{\pm\mathbf{.003}}$ | $0.975_{\pm.004}$ |
| $0.942_{\pm.010}$ | $0.205_{\pm.019}$ | $0.578_{\pm.015}$ | $\mathbf{0.492}_{\pm\mathbf{.008}}$ | $27.90_{\pm0.09}$ | $0.940_{\pm.004}$ | $\mathbf{0.969}_{\pm\mathbf{.009}}$ | $0.884_{\pm.005}$ | $0.975_{\pm.005}$ |
| $\mathbf{0.989}_{\pm\mathbf{.007}}$ | $0.216_{\pm.019}$ | $\mathbf{0.735}_{\pm\mathbf{.009}}$ | $0.501_{\pm.009}$ | $27.94_{\pm0.08}$ | $0.938_{\pm.004}$ | $0.956_{\pm.013}$ | $0.881_{\pm.006}$ | $\mathbf{0.997}_{\pm\mathbf{.001}}$ |
| $0.971_{\pm.014}$ | $0.209_{\pm.015}$ | $0.691_{\pm.010}$ | $0.532_{\pm.006}$ | $\mathbf{21.90}_{\pm\mathbf{0.23}}$ | $0.933_{\pm.004}$ | $0.941_{\pm.015}$ | $0.871_{\pm.008}$ | $0.983_{\pm.003}$ |

Table 10: Comparison between our method and StyleFlow [1] for sequential editing on FFHQ, where we edit each attribute of [age, yaw, glasses, smile] in a sequential order. Note that "w/o ss" means no subset selection, "w/o rw" means no reweighting of energy functions, and $\psi$ denotes the truncation coefficient of StyleGAN2.

| Methods | +age DES$_1\uparrow$ | ID$\downarrow$ | +yaw DES$_2\uparrow$ | ID$\downarrow$ | +glasses DES$_3\uparrow$ | ID$\downarrow$ |
|---|---|---|---|---|---|---|
| StyleFlow | $0.402_{\pm.003}$ | $0.329_{\pm.011}$ | $0.599_{\pm.010}$ | $\mathbf{0.187}_{\pm\mathbf{.004}}$ | $0.727_{\pm.032}$ | $\mathbf{0.187}_{\pm\mathbf{.006}}$ |
| LACE-ODE ($\psi$=0.5, w/o ss, w/o rw) | $0.491_{\pm.013}$ | $\mathbf{0.167}_{\pm\mathbf{.009}}$ | $0.498_{\pm.007}$ | $0.192_{\pm.011}$ | $0.889_{\pm.014}$ | $0.219_{\pm.009}$ |
| LACE-ODE ($\psi$=0.5, w/o rw) | $0.497_{\pm.012}$ | $\mathbf{0.167}_{\pm\mathbf{.009}}$ | $0.499_{\pm.014}$ | $0.192_{\pm.012}$ | $0.882_{\pm.016}$ | $0.220_{\pm.009}$ |
| LACE-ODE ($\psi$=0.5) | $\mathbf{0.558}_{\pm\mathbf{.009}}$ | $0.222_{\pm.011}$ | $\mathbf{0.693}_{\pm\mathbf{.011}}$ | $0.273_{\pm.016}$ | $\mathbf{1.003}_{\pm\mathbf{.010}}$ | $0.196_{\pm.012}$ |
| LACE-ODE ($\psi$=0.7) | $0.504_{\pm.005}$ | $0.254_{\pm.007}$ | $0.624_{\pm.014}$ | $0.281_{\pm.012}$ | $0.974_{\pm.011}$ | $0.198_{\pm.008}$ |

| +smile DES$_4\uparrow$ | ID$\downarrow$ | All DES$\uparrow$ | ID$\downarrow$ | FID$\downarrow$ | ACC$_y\uparrow$ | ACC$_s\uparrow$ | ACC$_a\uparrow$ | ACC$_g\uparrow$ |
|---|---|---|---|---|---|---|---|---|
| $0.533_{\pm.007}$ | $\mathbf{0.055}_{\pm\mathbf{.002}}$ | $0.565_{\pm.011}$ | $0.550_{\pm.010}$ | $44.02_{\pm1.45}$ | $0.821_{\pm.008}$ | $\mathbf{0.948}_{\pm\mathbf{.001}}$ | $0.870_{\pm.008}$ | $0.764_{\pm.012}$ |
| $0.800_{\pm.048}$ | $0.099_{\pm.003}$ | $0.669_{\pm.013}$ | $\mathbf{0.537}_{\pm\mathbf{.011}}$ | $27.33_{\pm0.04}$ | $\mathbf{0.910}_{\pm\mathbf{.006}}$ | $0.937_{\pm.003}$ | $\mathbf{0.995}_{\pm\mathbf{.002}}$ | $0.920_{\pm.006}$ |
| $0.787_{\pm.051}$ | $0.098_{\pm.004}$ | $0.666_{\pm.009}$ | $\mathbf{0.537}_{\pm\mathbf{.011}}$ | $27.28_{\pm0.16}$ | $\mathbf{0.910}_{\pm\mathbf{.007}}$ | $0.938_{\pm.003}$ | $\mathbf{0.995}_{\pm\mathbf{.002}}$ | $0.918_{\pm.005}$ |
| $\mathbf{0.929}_{\pm\mathbf{.030}}$ | $0.091_{\pm.007}$ | $\mathbf{0.796}_{\pm\mathbf{.007}}$ | $0.541_{\pm.010}$ | $27.22_{\pm0.22}$ | $0.905_{\pm.007}$ | $0.937_{\pm.003}$ | $0.992_{\pm.002}$ | $\mathbf{0.964}_{\pm\mathbf{.005}}$ |
| $0.853_{\pm.032}$ | $0.100_{\pm.007}$ | $0.739_{\pm.013}$ | $0.570_{\pm.013}$ | $\mathbf{21.76}_{\pm\mathbf{0.30}}$ | $0.897_{\pm.005}$ | $0.929_{\pm.003}$ | $0.980_{\pm.009}$ | $0.939_{\pm.001}$ |

{gender=female,smile=0,glasses=1,age=10} of FFHQ can be seen in Figure 13 and Figure 14, respectively.

### A.9.2 Compositions of energy functions

The 1024×1024 visual samples of our method (LACE-ODE) in compositions of energy functions with different logical operations: conjunction (AND), disjunction (OR), negation (NOT), and their recursive combinations on FFHQ can be seen in Figure 15 and Figure 16, respectively.

### A.10 Continuous control on discrete attributes

When the controlling attributes are discrete or binary, a smooth interpolation between discrete or binary attribute values could be a challenge of methods in controllable generation [10]. For example, can we smoothly control the amount of beard in the generated images even if its provided ground-truth labels are binary (0: without beard, 1: with beard)? To this end, we can add a temperature variable $T$ to the energy function of discrete attributes defined in Eq. (4), which becomes

$$E_\theta(c_i|x) = -\log \text{softmax}\left(\frac{f_i(x;\theta)[c_i]}{T}\right) := -\frac{f_i(x;\theta)[c_i]}{T} + \log \sum_{c_i} \exp\left(\frac{f_i(x;\theta)[c_i]}{T}\right)$$

where $c_i$ is a discrete attribute. Thus, the temperature $T \in (-\infty, 0)$ can be varied to adjust the impact of the attribute signal on the energy function. Its impact on the energy function becomes larger with a smaller value of $T$, resulting in a more significant visual appearance of the attribute value in the generated images, such as the increasing amount of beard on faces.

Table 11: Ablation results of LACE-euler (i.e., the Euler discretization method) with its hyperparameter "step_size" being set to 1e-2 or 1e-3. For notations, Infer – inference time (s: second), which refers to the single GPU time for generating a batch of 64 images.

| Methods | Infer↓ | FID↓ | ACC↑ |
|---|---|---|---|
| LACE-LD | 0.68s | **4.30** | 0.939 |
| LACE-euler (step_size=1e-2) | 0.68s | 6.31 | 0.969 |
| LACE-euler (step_size=1e-3) | 6.80s | 5.36 | 0.964 |
| LACE-ODE | **0.50s** | 6.63 | **0.972** |

In Figure 17, we show the 1024×1024 visual examples of continuous control on two binary attributes: (a) `beard` and (b) `smile` on the FFHQ data, where the visual appearance of both two attributes smoothly increases as we gradually decrease the temperature $T$.

### A.11 More results of ODE sampling vs. LD sampling

#### A.11.1 Hyperparameter settings

To compare the ODE and LD sampler more thoroughly, we perform a grid search in a large range of hyperparameter settings in each sampling method. In particular, The ODE sampler has two hyperparameters: (`atol`, `rtol`), which stand for the absolute and relative tolerances, respectively. The LD sampler has three hyperparameters: $(N, \eta, \sigma)$, which denote the number of steps, step size and standard deviation of the noise in LD, respectively.

For ODE, the grid search is performed with `atol` $\in$ [1e-1, 5e-2, 1e-2, 5e-3, 1e-3, 5e-4, 1e-4, 5e-5, 1e-5] and `rtol` $\in$ [1e-1, 5e-2, 1e-2, 5e-3, 1e-3, 5e-4, 1e-4, 5e-5, 1e-5]. Thus, there are 81 hyperparameter settings for the ODE sampler. For LD, the grid search is performed with $N \in [50, 100, 200, 300, 400, 500, 600, 1000]$, $\eta \in [0.1, 0.05, 0.01, 0.005, 0.001]$ and $\sigma \in [0.1, 0.05, \eta]$. Thus, there are 104 hyperparameter settings for the LD sampler.

#### A.11.2 Impact of each individual hyperparameter

To dissect how sensitive the samplers are to each individual hyperparameter, Figure 18 shows the impact of (`atol`, `rtol`) in the ODE sampler (top row) and the impact of $(N, \eta, \sigma)$ in the LD sampler. We can see that with different `rtol` values, a smaller `atol` tends to have a higher ACC score (though it slightly decreases after `atol` $< 10^{-3}$) and a lower FID score. Hence, we could always use small values of (`atol`, `rtol`) to get both good generation quality and controllability, which implies the hyperparameters in the ODE sampler are easy to tune.

In the LD sampler, however, there exists a clear ACC-FID trade-off controlled by the standard deviation of the noise $\sigma$: a smaller value of $\sigma$ results in a better ACC score but a worse FID score. Meanwhile, increasing the number of steps $N$ will also cause a better ACC score but a worse FID score when the value of $\sigma$ is small. Therefore, both values of $N$ and $\sigma$ in the LD sampler should not be too large or too small, and a sweet pot of these hyperparameters varies with different downstream tasks as we see in our experiments, which implies it tends to be more difficult to find the optimal hyperparameter setting for the LD sampler.

#### A.11.3 Ablation on a simple Euler method

By default, LACE-ODE applies the adaptive-step "dopri5" solver (i.e., Runge-Kutta of order 5) because of its adaptivity in step size for better efficiency. But how does our ODE formulation in Eq. (7) work with a simple Euler discretization method? To this end, we run our ODE sampler with the Euler method (called LACE-euler) on CIFAR-10, with an extra hyperparameter "step_size" being set to 1e-2 or 1e-3. The results are shown in Table 11. We can see that 1) LACE-ODE with the default "dopri5" method is faster than LACE-euler (0.50s vs 0.68s) for getting similar performance, which confirms our intuition of adaptive step size vs. fixed step size, and 2) our method also works decently well with the Euler method, and its performance lies in-between that of LACE-LD and LACE-ODE.

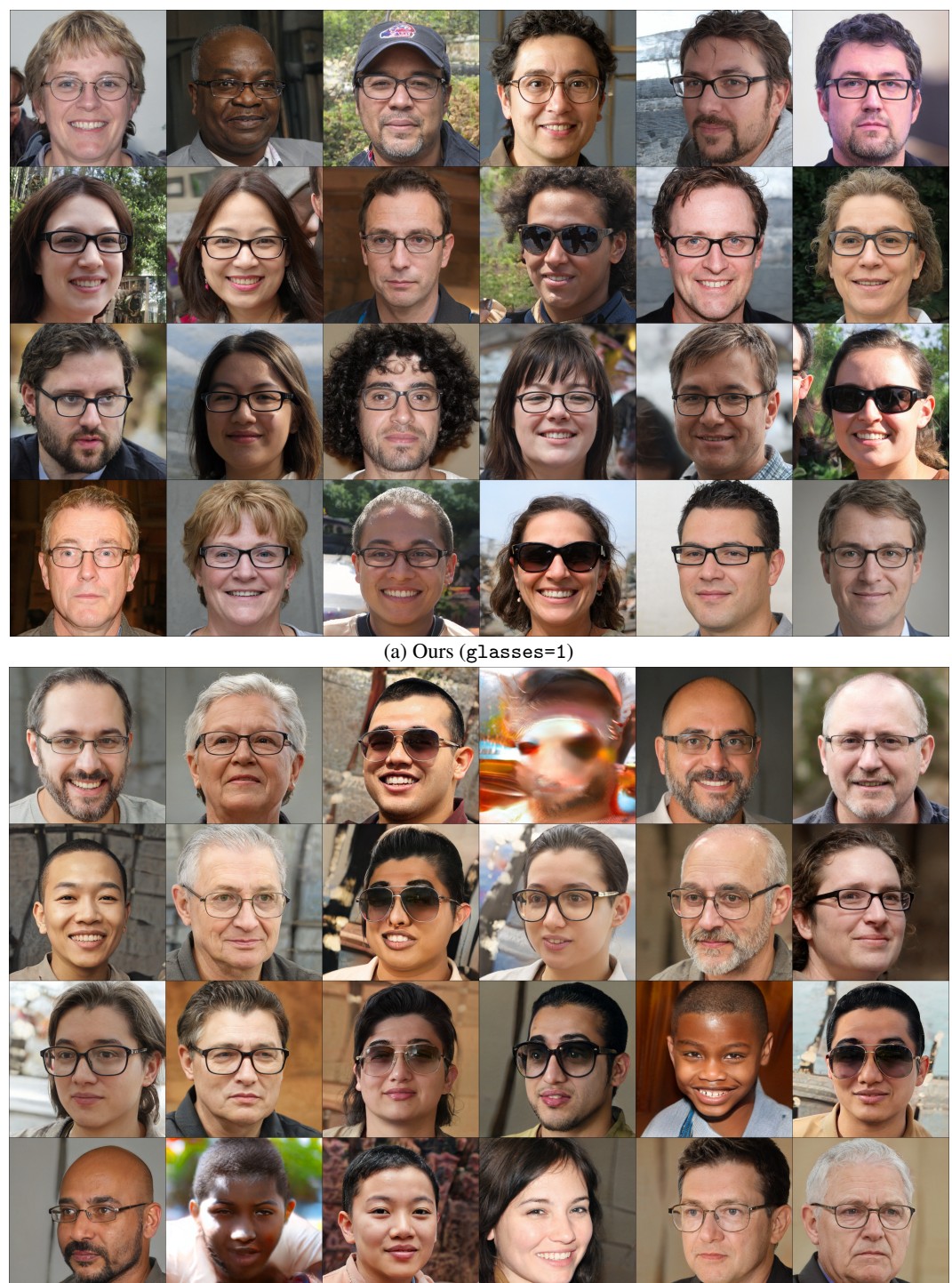

(a) Ours (`glasses=1`)

(b) StyleFlow (`glasses=1`)

Figure 10: Uncurated 1024×1024 conditional sampling results of our method (LACE-ODE) and StyleFlow on {`glasses=1`} of FFHQ, where our method outperforms StyleFlow in terms of image quality (or diversity) and controllability.

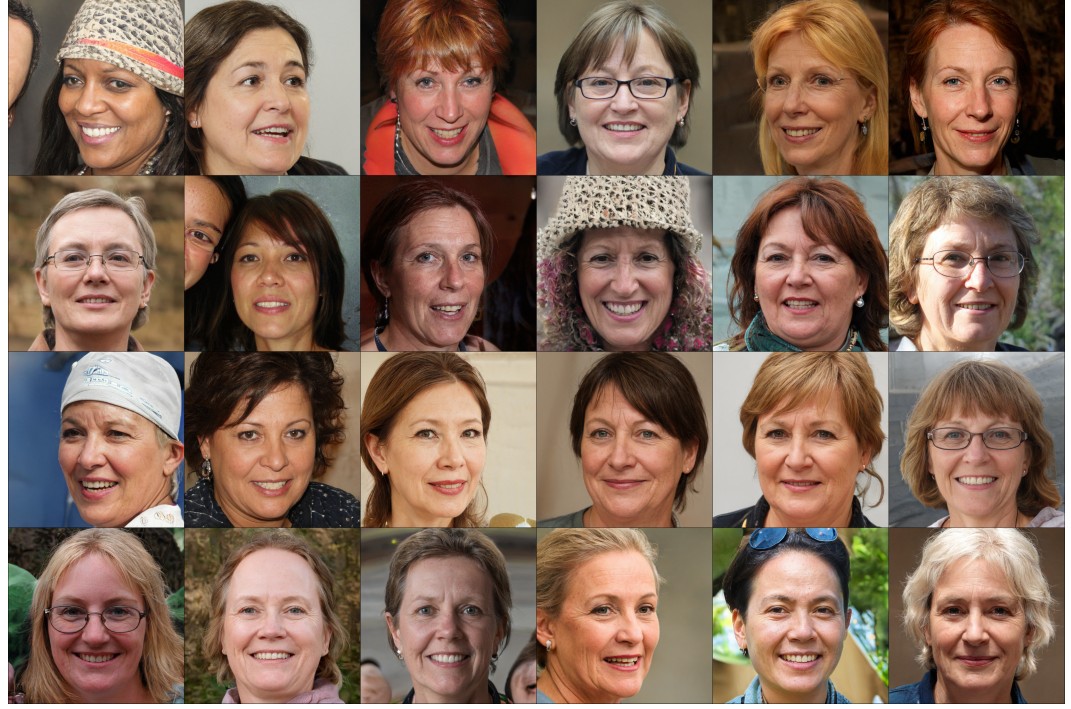

(a) Ours (`gender=female,smile=1,age=55`)

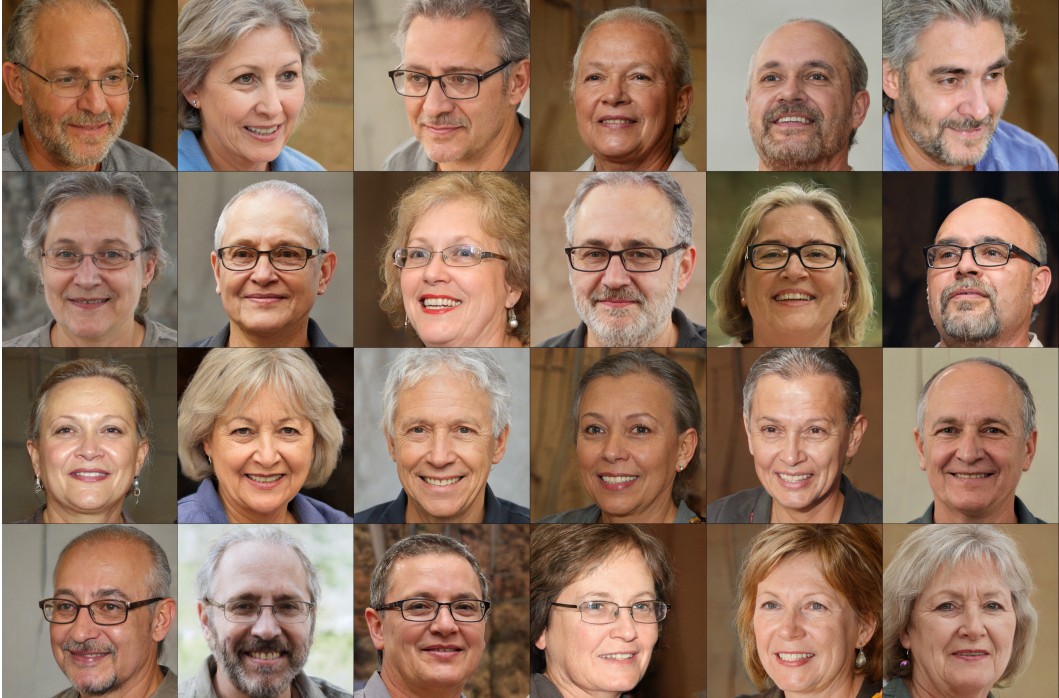

(b) StyleFlow (`gender=female,smile=1,age=55`)

Figure 11: Uncurated 1024×1024 conditional sampling results of our method (LACE-ODE) and StyleFlow on {`gender=female,smile=1,age=55`} of FFHQ, where our method outperforms StyleFlow in terms of image quality (or diversity) and controllability.

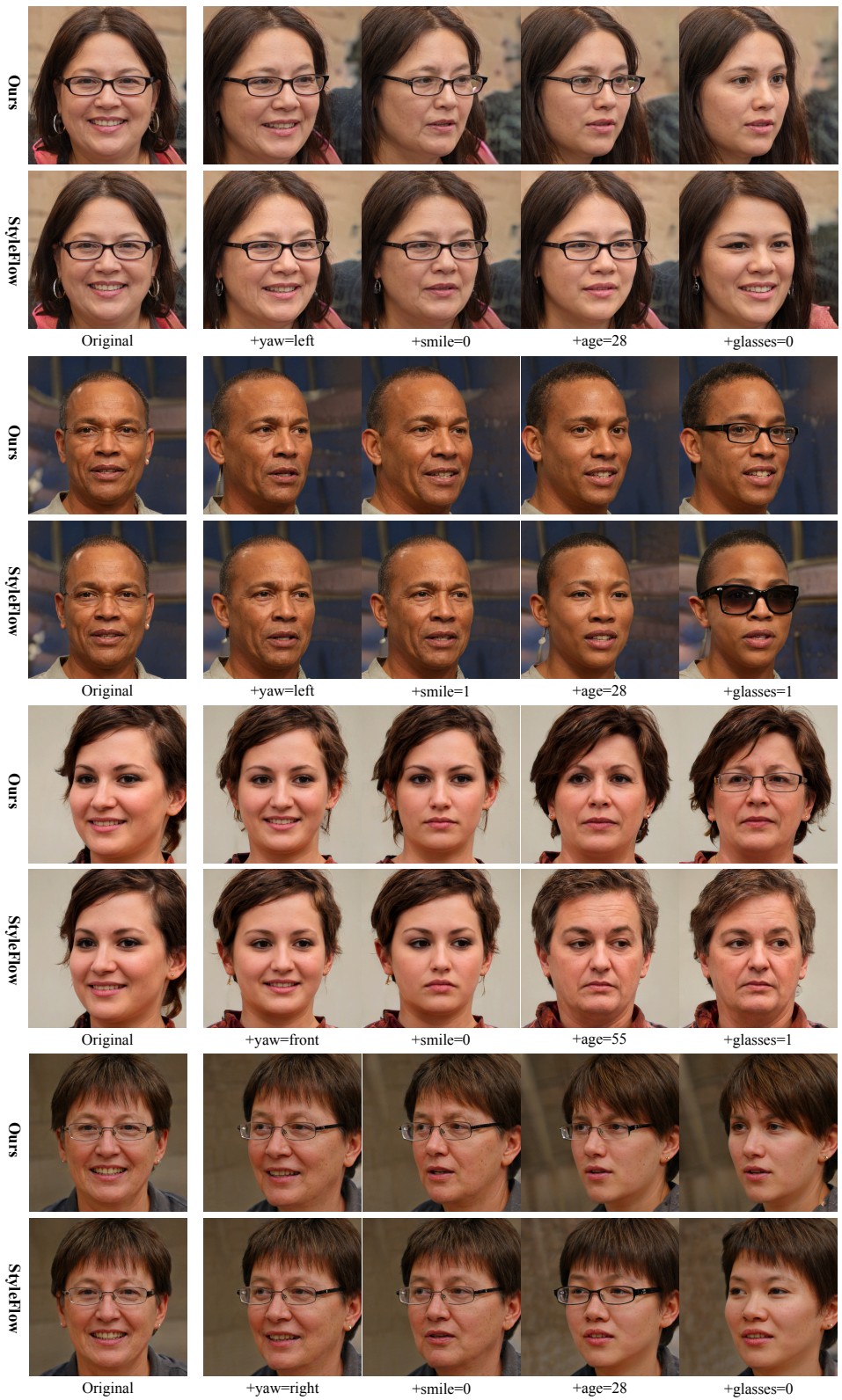

Figure 12: The sequential editing results of our method (LACE-ODE) and StyleFlow on FFHQ with a sequence of [yaw,smile,age,glasses]. Note that '+' means the current editing is built upon the last edited images. For instance, +smile=0 refers to changing the last edited images to make them not smile. Overall, compared to StyleFlow, our method can successfully perform each edit while less affecting other attributes and face identities.

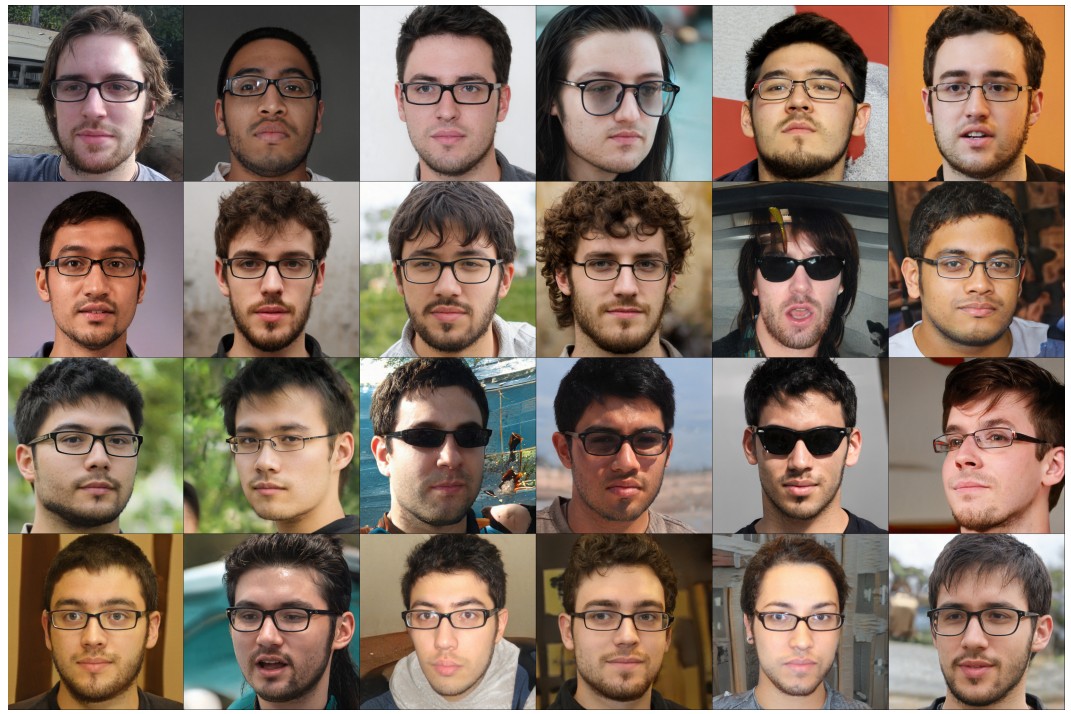

(a) Ours (`beard=1,smile=0,glasses=1,age=15`)

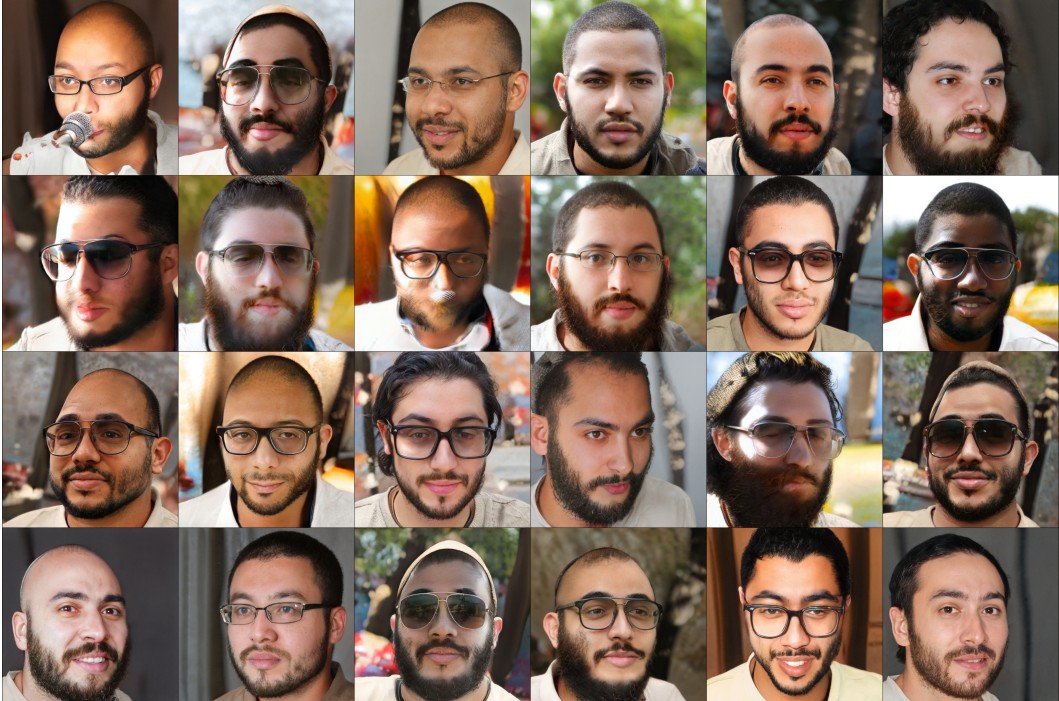

(b) StyleFlow (`beard=1,smile=0,glasses=1,age=15`)

Figure 13: Uncurated 1024×1024 zero-shot generation results of our method (LACE-ODE) and StyleFlow on the unseen attribute combinations {`beard=1,smile=0,glasses=1,age=15`} of FFHQ, where where our method excels at zero-shot generation while StyleFlow performs significantly worse by either generating low-quality images or completely missing the conditional information.

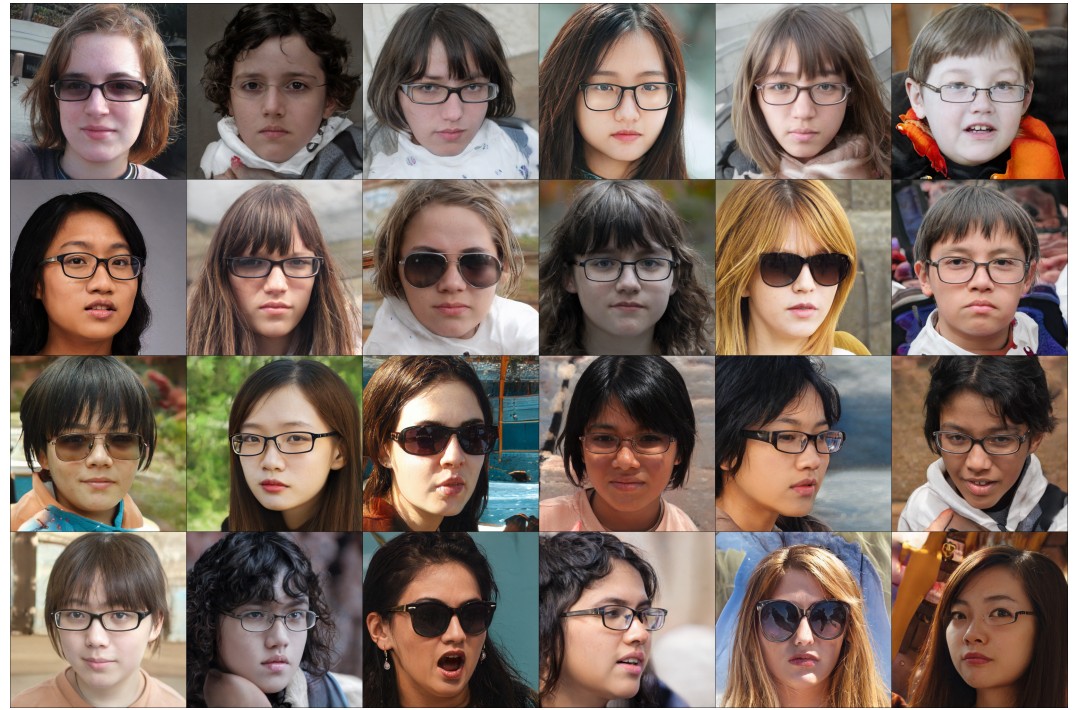

(a) Ours (`gender=female,smile=0,glasses=1,age=10`)

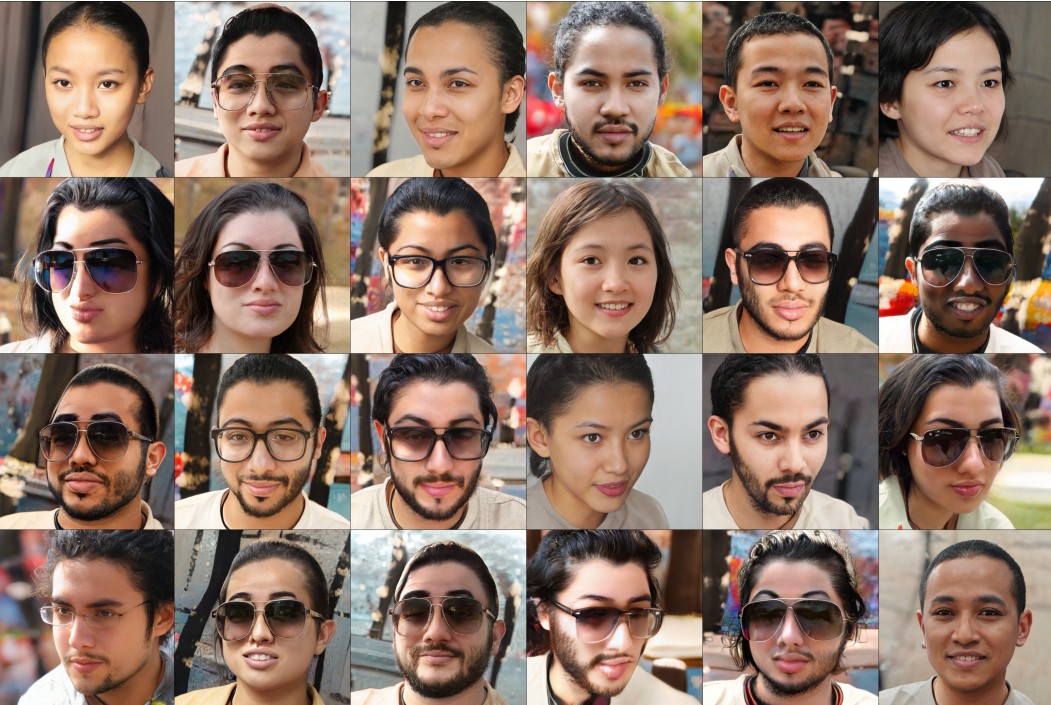

(b) StyleFlow (`gender=female,smile=0,glasses=1,age=10`)

Figure 14: Uncurated 1024×1024 zero-shot generation results of our method (LACE-ODE) and StyleFlow on unseen attribute combinations {`gender=female,smile=0,glasses=1,age=10`} of FFHQ, where our method excels at zero-shot generation while StyleFlow performs significantly worse by either generating low-quality images or completely missing the conditional information.

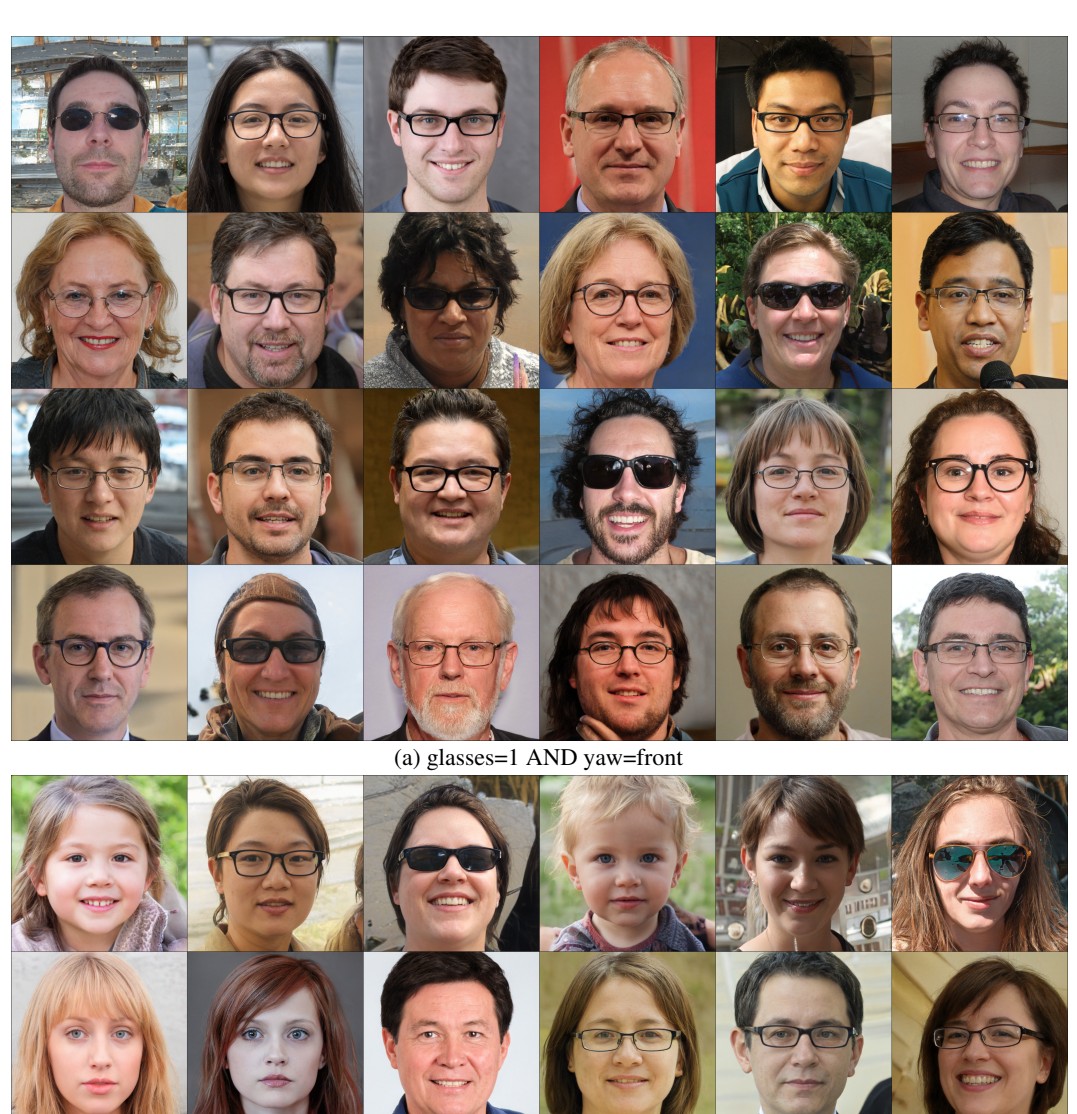

(a) glasses=1 AND yaw=front

(b) glasses=1 OR yaw=front

Figure 15: Uncurated 1024×1024 generation results of our method (LACE-ODE) in compositions of energy functions with different logical operations: (a) conjunction (AND), and (b) disjunction (OR). We can see that our method closely follows the rule of the given logical operators, and also produces diverse images that cover all possible logical cases.

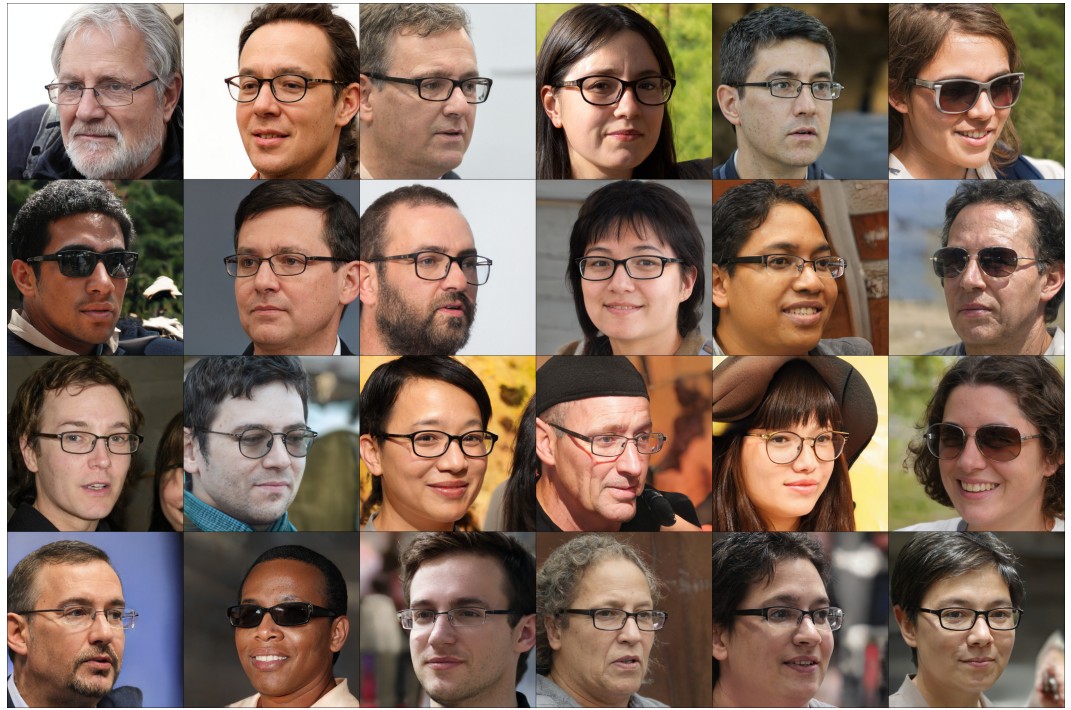

(a) glasses=1 AND (NOT yaw=front)

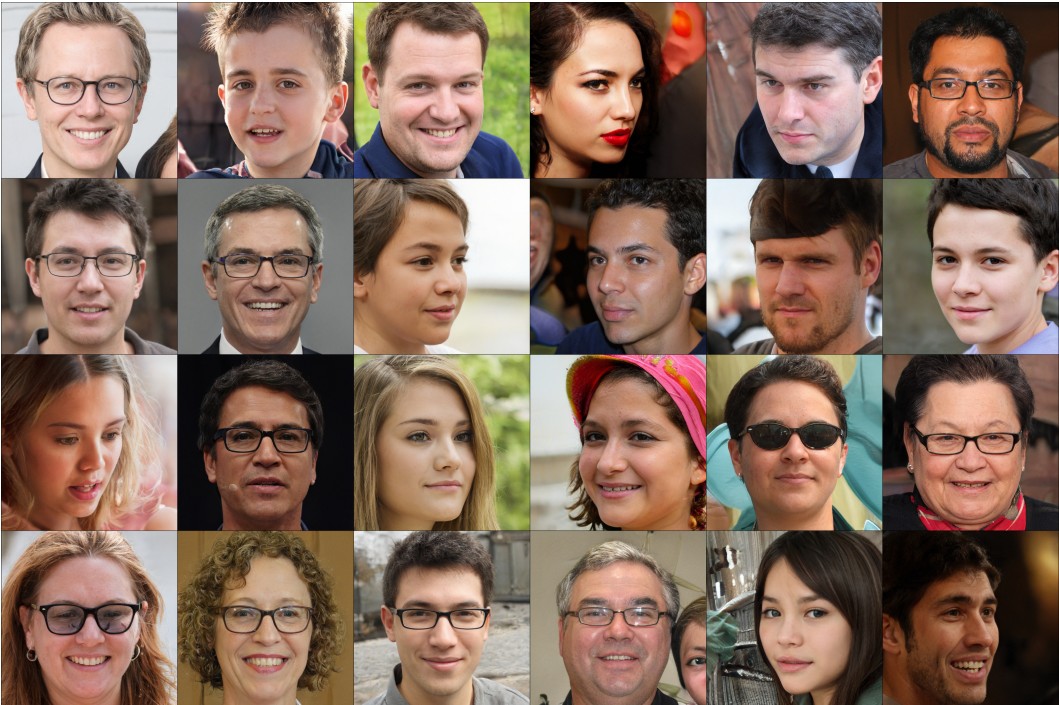

(b) (glasses=1 AND yaw=front) OR (glasses=0 AND (NOT yaw=front))

Figure 16: Uncurated 1024×1024 generation results of our method (LACE-ODE) in compositions of energy functions with different logical operations: (a) negation (NOT), and (b) recursive combinations of logical operations on FFHQ. We can see that our method closely follows the rule of the given logical operators, and also produces diverse images that cover all possible logical cases.

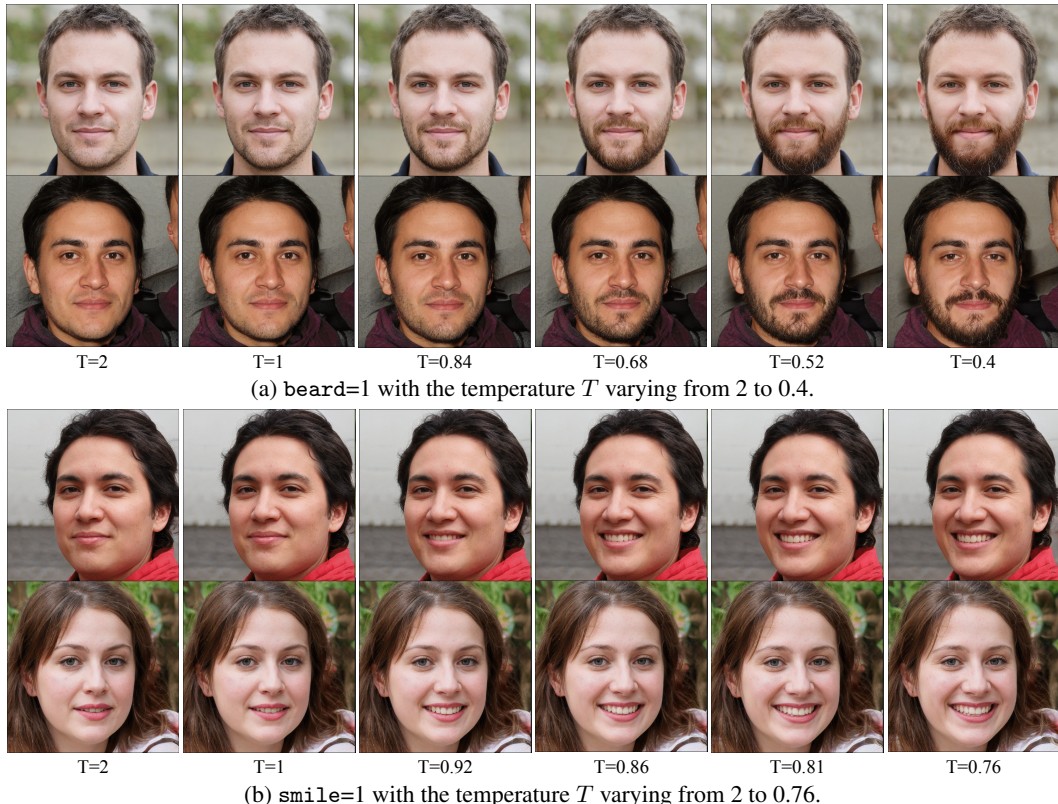

(a) `beard=1` with the temperature $T$ varying from 2 to 0.4.

(b) `smile=1` with the temperature $T$ varying from 2 to 0.76.

Figure 17: Continuous control of discrete attributes (a) `beard` and (b) `smile` by varying the temperature $T$. Even if the ground-truth labels of the above discrete attributes are binary only, our method can learn to smoothly perform continuous control of them. As we decrease the temperature $T$, the visual appearance of the controlling attributes becomes more significant in the generated images.

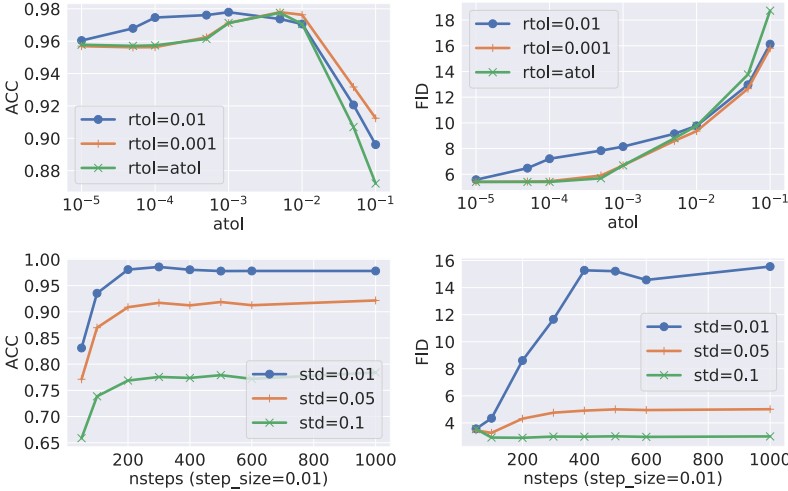

Figure 18: The impact of hyperparameters (top: ODE, bottom: LD). We can see that we could always use small values of (`atol`, `rtol`) to get both good generation quality and controllability in the ODE sampler, which implies the hyperparameters in the ODE sampler are easy to tune. On the contrary, there exists a clear ACC-FID trade-off controlled by the standard deviation of the noise $\sigma$ and the number of steps $N$, which implies it tends to be more difficult to find the optimal hyperparameter setting for the LD sampler.