# OpenReview forum: "Controllable and Compositional Generation with Latent-Space Energy-Based Models"
_NeurIPS.cc/2021/Conference — NeurIPS 2021 Poster_

### Official Review · Reviewer_K3yB · 2021-07-02

**Rating:** 6
**Confidence:** 4

**Summary:**

The paper proposes an effective and efficient approach to conditional image generation using GANs. Two main advances are described 1) The use of EBM in latent space that don’t require the retraining of image generators. This allows for very efficient training since the generator can be held fixed. 2) The use of an ODE solver for sampling that is less sensitive to hyperparameters than traditional LD solvers.

Results are shown on CIFAR-10 and FFHQ datasets. This includes results on sequential generation, compositional generation and zero-shot generation.


**Ethical Concerns:**

None.

**Limitations And Societal Impact:**

Limitations and societal impact are well-written. It would be good to cover in societal impact the implications of conditional GANs vs. unconditioned GANs. I.e., conditioned GANs may be more useful for nefarious applications.

**Main Review:**

Overall this paper is borderline. The improvements in efficiency and the novelty of the approach push it slightly over the top, but the limitations in the experimental results keeps the paper from being a clear accept.

Positives:

++ EBM in latent space leads to very efficient training.

\+ Results show good disentanglement of attributes.

\+ Both ODE and LD solutions provide good results.

\+ Results are impressive with respect to StyleFlow

\+ Paper is well-written and easy to follow.

Negatives:

\- Inference time is longer than other approaches, but still not excessive.

\- The experimental set-up was varied from StyleGan paper, so hard to compare results.

\- It is unclear how inference time scales with the number of conditioned attributes.

\- No experiments on other high-res datasets except FFHQ.

\- Results on FFHQ only compare against StyleFlow. The StyleFlow paper offers comparisons to three other methods. This might be okay if the experimental set-up was the same, but it is not.

\- Results for sequential generation only use a single ordering. Are similar results found with different orderings? Be good to show a figure similar to Fig. 7 in StyleFlow paper.

Other comments:

+ The compositions of energy functions section (Line 239) doesn’t add much to the paper. It’d be better to use this space to compare against more approaches.

+ Results missing in several figures. Figure 2a left, Figure 4b left

+ Since you’re generating 1024x1024 images, it’d be good to show high-res crops (zoom-ins) showing the details of the images.


**Time Spent Reviewing:**

3 hours

---

> ### Author Response · Authors · 2021-08-10
> **Responses to Reviewer K3yB**
>
> We thank the reviewer for the thoughtful feedback, and we provide extra experimental results to clarify the concerns
> that the reviewer had.
>
> **Q:** Inference time is longer than other approaches, but still not excessive.
>
> Good point! **Our inference time can be further reduced without sacrificing the performance**. In our current setting, each attribute classifier is parametrized by a separate (384-256-128) MLP network
> (see Table 4 in the Appendix). That is, when conditioning on $n$ attributes, we have $n$ separate MLP networks. We found
> that the inference time of our method largely depends on the number of MLP networks. Accordingly, if we use a single MLP
> network with the same size and multiple prediction heads, each of which corresponds to one attribute, we can further
> reduce the inference time without sacrificing the controllable generation performance. Please see the quantitative
> results below.
>
> **Q:** It is unclear how inference time scales with the number of conditioned attributes.
>
> We thank the reviewer for the suggestion. As the reviewer suggested, **we ran our method for conditional sampling with the
> increasing number of attributes (1-5)**. Without loss of generality, we consider the test case: “glasses” (1), “age,
> glasses” (2), “smile, age, glasses” (3), “gender, smile, age, glasses” (4), “yaw, gender, smile, age, glasses” (5). The
> sampling time for different numbers of attributes is listed in the table. Note that “separate MLPs” denotes the current
> setting where we use $n$ separate MLP networks for $n$ attributes, and “a single MLP” denotes the new setting where we
> use a single MLP network with the same size and $n$ prediction heads for $n$ attributes. We can see that although the
> inference time increases with the number of attributes in both cases, **the new setting (“a single MLP”) has much smaller
> inference time, and the advantage becomes larger with more attributes**.
>
> |                               | #attributes=1  | #attributes=2  | #attributes=3  | #attributes=4  | #attributes=5 |
> | :---:                         |    :----:      |      :---:     | :---:          | :---:          | :---:         |
> | Inference time (separate MLPs)| 0.68           | 2.34          | 4.36            | 6.65           | 7.84          |
> | Inference time (a single MLP) | 0.68           | 1.70          | 2.25            | 2.58           | 2.63          |
>
> Meanwhile, **the performances remain similar**. For instance, in the case of conditioning “yaw, gender, smile, age,
> glasses” (5), the ACCs of “separate MLPs” and “a single MLP” are shown as follows:
>
> |                               | ACC_yaw        | ACC_gender     | ACC_smile      | ACC_age        | ACC_glasses   |
> | :---:                         |    :----:      |      :---:     | :---:          | :---:          | :---:         |
> | separate MLPs | 0.927          | 0.956          | 0.953          | 0.897          | 0.994         |
> | a single MLP    | 0.904          | 0.973          | 0.954          | 0.892          | 0.984         |
>
> We will add the results to the revised paper, as an alternative to reduce the sampling time.
>
> **Q:** The experimental set-up was varied from StyleGan paper, so hard to compare results.
>
> We agree that our experimental settings are slightly different from StyleGAN, mainly because StyleGAN does not measure
> accuracy in conditional generation while our goal is to achieve both high accuracy and high sampling quality together.
> For the results on CIFAR-10, the FID evaluation setting is almost the same with StyleGAN so the FID scores are
> comparable to the StyleGAN paper. For the results on FFHQ, we are following the experimental protocol used in
> controllable generation literature (specifically StyleFlow which obtains state-of-the-art results currently). If there
> is any particular comparison against StyleGAN that the reviewer is interested in, we can do during the discussion
> period.
>
> **Q:** No experiments on other high-res datasets except FFHQ.
>
> We recently ran our method on **two new datasets: MetFaces (1024x1024) and AFHQ (512x512)** with the pre-trained
> StyleGAN2-ADA model for controllable generation. The results are also very similar to FFHQ: Our method can effectively
> control various attributes and their combinations while preserving the high image quality. Please see the anonymous link
> to the corresponding visual examples: https://anonymous4lace.github.io/lace/. We will add these new results to the
> revised paper.
>
> **Q:** Results on FFHQ only compare against StyleFlow.
>
> On one hand, we agree with the reviewer that for sequential editing on FFHQ, we only compared with StyleFlow because it
> is the only state-of-the-art baseline in this task that we found. The StyleFlow paper showed that it clearly outperforms
> their baselines (Image2StyleGAN, InterfaceGAN, GANSpace) regarding both image and editing quality. As we closely follow
> the StyleFlow setting and our method largely outperforms StyleFlow, we think it is a little redundant to compare with
> these three baselines (Image2StyleGAN, InterfaceGAN, GANSpace).
>
> On the other hand, for conditional sampling on FFHQ, we did compare with several baselines (including StyleFlow and
> Latent-JEM). We also want to emphasize that many of our related methods are only capable of or have only shown the
> *conditional generation* results for low-resolution images (i.e., CIFAR-10), such as JEM, Cond-EBM, VP-SDE, and VE-SDE.
> This is the reason why we compared with many more baselines on CIFAR-10 than FFHQ.
>
> **Q:** Results for sequential generation only use a single ordering.
>
> We thank the reviewer for the suggestion. As the reviewer suggested, **we ran our method for sequential editing with
> different orderings**. We first consider a sequence of [“glasses”, “age”, “gender”, “smile”, “yaw”] by reversing the
> original ordering in our paper, and the results after all edits are shown as follows:
>
> |  Methods  | DES        | ID     | FID      | ACC_glasses    | ACC_age   | ACC_gender   | ACC_smile   | ACC_yaw  |
> | :---:     |    :----:  |  :---:     | :---:     | :---:  | :---:   | :---:  | :---:  | :---:  |
> | StyleFlow | 0.542    | 0.617      | 44.74      | 0.880     | 0.806     | 0.722     | 0.753     | **0.946**   |
> | LACE-ODE  | **0.664**  | **0.606** | **28.14** | **0.998** | **0.896** | **0.938** | **0.967** | 0.914   |
>
> Moreover, we consider a sequence of [“age”, “yaw”, “glasses”, “smile”] by changing the number of attributes and randomly
> perturbing the ordering, and the results after all edits are shown as follows:
>
> |  Methods  | DES        | ID     | FID      | ACC_age    | ACC_yaw   | ACC_glasses   | ACC_smile   |
> | :---:     |    :----:  |  :---:     | :---:     | :---:  | :---:   | :---:  | :---:  |
> | StyleFlow | 0.565    | 0.550      | 44.02      | 0.821     | **0.948** | 0.870     | 0.764     |
> | LACE-ODE  | **0.669**  | **0.537** | **27.33** | **0.910** | 0.937     | **0.995** | **0.920** |
>
> We can see that the experimental results are consistent with different attribute orderings: **Our method largely
> outperforms the state-of-the-art StyleFlow in various sequential editing tasks**.
>
> **Final remark**: Thank you for providing constructive and detailed feedback on our submission. If you have any
> additional questions/comments/concerns, please feel free to let us know here. **Otherwise, we would appreciate it if you
> consider raising your rating of this submission.**

---

### Official Review · Reviewer_vR7r · 2021-07-13

**Rating:** 6
**Confidence:** 4

**Summary:**

The paper considers the task of obtaining additional control over a pre-trained generative model without re-training it. Specifically, it aims to generate conditional samples from an unconditional StyleGAN by only training a classifier with respect to the conditioning information. To achieve this, it considers the conditional distribution of StyleGAN latent codes. By Bayes Theorem, this conditional distribution is proportional to the unconditional distribution of the latent codes (which has a known standard normal distribution) times the distribution of conditioning information given the corresponding sample. The latter is modeled as a Gibbs distribution with an energy derived from the classifier. Two approaches to sample from this unnormalized distribution are evaluated: (i) Langevin sampling and (ii) Performing gradient descent on the classifier-derived energy. Experiments demonstrate good performance of this approach with significant improvements over StyleFlow for combinations of conditionings which are not in the training data.

**Limitations And Societal Impact:**

Limitations regarding the need for powerful pre-trained generative models and the availability of labels to train the classifier are addressed and general potential impacts of generative models which are inherited by the presented approach are mentioned.

**Main Review:**

**Strengths**
- The paper is well motivated by the desire to reuse powerful generative models without having to retrain them, which is quite costly. In particular, it is difficult to anticipate what needs regarding control over a generative model are desired by users and it is therefore necessary to develop techniques that enable customized controls in a computationally efficient post-hoc manner.
- The approach is simple and shows good performance. Based on this it could serve as an important baseline for face editing works and enjoy wide adaption.
- Besides good quality of the synthesized results, it achieves the goal of providing flexible control with little computational requirements as demonstrated by comparatively short training and inference times.
- Especially in the case of rare combinations of conditioning information, the approach demonstrates improved performance over StyleFlow. In general, the formulation of controlled synthesis as a composition of different energies
provides a flexible control mechanism.

**Weaknesses**
- Both the idea that classifiers can be interpreted as (part of) an EBM (Grathwohl, Will et al. “Your Classifier is Secretly an Energy Based Model and You Should Treat it Like One.” 2020) as well as the idea that optimizing latent codes of generators instead of pixels improves the quality when optimizing an energy derived from classification networks (Nguyen, Anh M et al. “Synthesizing the preferred inputs for neurons in neural networks via deep generator networks.” 2016) are known. Similarly, the ODE formulation follows [38] (although I'm not really sure why that derivation is necessary to formulate Eq. (10); see points below) and the composition of energies follows [8]. Thus, the main contribution is the evaluation of these techniques in a very specific setting.
- The paper makes the very restrictive assumption that the latent distribution follows a standard normal distribution. Thus, it assumes that the most difficult part regarding generative modeling is already solved. Moreover, it considers only relatively easy datasets with CIFAR-10 and FFHQ. To quote from Upchurch, P. et al. “Deep Feature Interpolation for Image Content Changes.” 2017: "Generative models can be much more powerful than linear interpolation, but the current problems (in particular, face attribute editing) which are used to showcase generative approaches are too simple." Here, the situation is even more severe because it makes the problem even simpler by relying on StyleGAN, such that one is directly working with a normally distributed, relatively low-dimensional code. Note that for example Nguyen 2016 does not make assumptions regarding normally distributed codes and considers more complex datasets such as ImageNet and MIT Scenes.
- While it is always interesting to see connections between different approaches and works, I fail to see how the extensive discussion in Sec. 2 of energy based models, langevin sampling and stochastic differential equations is necessary to arrive at Eq. (10). The reverse ODE to Eq. (10) describes gradient flow on the latent code with respect to a classification/regression network. Optimizing inputs with respect to such a network is a very natural idea that has been considered many times in different contexts in the literature (like Nguyen 2016 and Upchurch 2017 but also Goetschalckx, Lore et al. “GANalyze: Toward Visual Definitions of Cognitive Image Properties.” 2019 and similar works on finding interpretable directions in the latent space of GANs, encoding images into this latent space or performing superresolution as in Menon, Sachit et al. “PULSE: Self-Supervised Photo Upsampling via Latent Space Exploration of Generative Models.” 2020). This discussion might be useful if it would unify previous approaches but in its current form it only seems to present a simple approach in an unnecessarily complex way.
- Regarding the derivation of Eq. (10): For $f(x,t)=-\frac{1}{2}\beta(t)x$ and $g(t)=\sqrt{\beta(t)}$ (defined in l.137), the reverse SDE according to Eq. (6) in [38] should read $dx = -\frac{1}{2}\beta(t) \[ x + 2 \nabla_x \log p_t(x) \]dt + \sqrt{\beta(t)} d\bar{w}$. Eq. (7) in the paper under review does not contain the factor 2 on the score $\nabla_x \log p_t(x)$. Because of this, $z$ and $\nabla_z \log p_t(z)$ in Eq. (9) cancel under the assumption that $p_t$ is standard normal. However, with the factor of 2, these terms would not cancel and Eq. (10) would include an additional term $\frac{1}{2}\beta(t) z dt$. Intuitively this also makes more sense to me since the latter would account for the fact that the marginal distribution of $z$ is standard normal. In this case, the resulting (reverse) ODE would also directly correspond to the gradient flow on the energy in Eq. (6). So which version is correct and which one was used in the implementation?
- One differences to other approaches that optimize latent codes is the use of the ODE solver [5] to solve (the reverse) of the gradient flow in Eq. (10) instead of using a simple Euler discretization commonly used for gradient descent. However, there is no ablation regarding this choice. Is it important to use [5] or does a simple Euler discretization work just as well? Similarly, the choice of the time-dependent diffusion coefficient beta is never discussed. What is used for beta? What is the effect of it?

**Comments and Suggestions**
- I think the paper demonstrates precisely the fact that face attribute editing with the help of pre-trained generative models is too simple: The final "model" that is proposed by this work is to optimize latent codes with respect to a desired output from a classification or regression model. Given the good results obtained by this approach is a good indication that this field (face attribute editing) should reconsider its baselines. Thus, I think the paper could become a valuable contribution by pointing this out and presenting its simple and strong approach that can serve as a baseline for future works. I think it is very important to have simple and strong approaches but unfortunately, in its current form, the paper seems to try to hide the simplicity of its approach.
- Why not formulate directly the distribution of x/z conditioned on c instead of talking about the joint distribution most of the time when the goal is to sample the former?
- I think the results from the ablation regarding which space to train the classifier on in A.4 are quite interesting and important. This choice seems to have a significant effect on the performance, and I guess it is also the reason for the short training time? How does training time on w space compare to training time on image space?

**Rating**
- The paper presents a simple approach to utilize a pretrained, unconditional GAN for conditional sampling and demonstrates good results. While it could serve as a simple and strong baseline for face attribute editing, in its current form the paper presents a different message regarding a "novel formulation of a joint EBM". Given that the model performs gradient descent on the latent code of StyleGAN this seems like an overclaim and I fail to see the novelty and contribution. The paper must reconsider to what purpose it draws the connection to energy based models. In its current form, this connection is not very natural and rather seems to, unnecessarily, hide a simple approach in a complex derivation.
- Thus, in my preliminary rating I do not recommend accepting the paper, but I would like to hear why the authors think that the connection to energy based models is important (especially with respect to Eq. (10)) and how this gives the paper more significance than directly stating that a simple baseline which performs gradient descent with respect to a classifer in the latent space of StyleGAN outperforms recently proposed and more complex approaches like StyleFlow.

**Time Spent Reviewing:**

16

---

> ### Author Response · Authors · 2021-08-10
> **Responses to Reviewer vR7r**
>
> We thank the reviewer for the detailed and thoughtful feedback. **Please note that there is a typo (i.e., a missing constant
> coefficient) in Eq. (7) and the ODE formulation in Eq. (10) is correct after fixing the typo.** In the following, we
> provide a detailed clarification on our contribution and on the misunderstandings that the typo may have caused.
>
> **Q:** The idea that classifiers can be interpreted as (part of) an EBM is known.
>
> We do not claim “classifiers can be interpreted as (part of) an EBM” is one of our contributions. The JEM paper
> (Grathwohl et al. 2020) need to train both the generative model $p(x)$ and the classifier $p(c|x)$. Instead, by
> formulating EBM in latent space, we only need to train the classifier for controllable generation. This difference makes
> our method much simpler and faster in training than JEM (Grathwohl et al. 2020).
>
> **Q:** The idea that optimizing latent codes of generators instead of pixels improves the quality is known.
>
> Thanks for pointing out this connection. We agree that Nguyen et al. 2016 shares the similar idea of optimizing latent
> variables of generators with an image classifier for image synthesis. But there are two major differences: 1) **Nguyen
> et al. 2016 can be approximately treated as a special case of our method**, where their classifier is defined in the
> pixel space. Due to the reparameterization trick, the classifier in our method can be defined in various spaces (
> pixel/w/z) of a generator, which gives additional flexibility to our model. In fact, our experiments show that w-space
> actually works best, which also explains why Nguyen et al. 2016 has poor image quality. 2) Nguyen et al. 2016 proposed
> the idea with the intuition of using generative models as a powerful prior for activation maximization, while **our
> method is formulated in a principled way from the EBM perspective**. We will be happy to include this discussion in our
> paper.
>
> **Q:** The ODE formulation follows [38].
>
> Although our ODE formulation follows [38], we have extensively discussed its difference with [38] (mainly lines
> 142-144): **By formulating EBM in the latent space, we only need to train a *time-invariant* classifier and we do need
> to train the score function**, making the training much simpler and the inference much more efficient than [38]. No
> prior work has used ODE sampling in the context of explicit EBMs, and we found the ODE sampling is faster and more
> robust to hyperparameters than the de facto LD sampling for EBMs.
>
> **Q:** The composition of energies follows [8].
>
> Although our “compositions of energies” is inspired by [8], there are two major differences: 1) **[8] applied logical
> operations to *conditional* energy functions $E(x|c_i)$ which requires learning an EBM generator in the pixel space**.
> Unfortunately, learning EBMs in the data space using MCMC sampling is very challenging. In contrast, we compose *joint*
> energy functions $E(z, c_i)$ in the latent space, which is simpler, faster in both training and inference. 2) Compared
> to [8], **our synthesized images have much better visual quality and higher resolution**.
>
> **Q:** The paper makes the very restrictive assumption that the latent distribution follows a standard normal
> distribution.
>
> There might be some confusion here. In most of the commonly-used generative models (such as GANs, VAEs, Normalizing
> Flows, and even score-based models etc.), the latent prior distribution is a standard Gaussian. Please note that even in
> VAEs with hierarchical priors, the prior can be converted into a standard Gaussian prior as discussed in Sec. E in [a].
> This is not restrictive as we can obtain a highly expressive generative model for controllable generation.
>
> [a] Vahdat et al. Score-based Generative Modeling in Latent Space, 2021.
>
> **Q:** It assumes that the most difficult part regarding generative modeling is already solved. Moreover, it considers
> only relatively easy datasets with CIFAR-10 and FFHQ.
>
> For practical purposes, we model the generation of high-quality images using a pre-trained GAN and we introduce EBMs for
> controllability and compositionality. More recently, we experimented our method on **two new datasets: MetFaces
> (1024x1024) and AFHQ (512x512)**, and also got very good results of controllable generation. Please see the anonymous link
> to the corresponding visual examples: https://anonymous4lace.github.io/lace/. We will update our paper with the new
> results.
>
> **Q:** Linear interpolation (in particular, face attribute editing) is too simple to show the power of generative
> models.
>
> We agree that only considering *linear* interpolation in face attribute editing may be a little simple, but our settings
> are different as follows: 1) Optimizing latent space with our EBM formulation is a highly *nonlinear* process. 2)
> Sequential editing with long sequences is more difficult than editing one single attribute. 3) Compositional
> generalization, the main goal of our work, is clearly more challenging than attribute editing.
>
> **Q:** I fail to see how to arrive at Eq. (10). It only seems to present a simple approach in an unnecessarily complex
> way.
>
> **The ODE formulation in Eq. (10) is correct, and we thank the reviewer for pointing out a typo in Eq. (7)**. Specifically,
> there is indeed a factor 2 on the score $\nabla_x \log p_t(x,c)$. Only by fixing the typo, we can correctly derive Eq.
> (8) from Eq. (7), where the idea is to use the Fokker-Planck equation (Øksendal, 2003) to transform an SDE to an ODE
> (Appendix D.1 in [38]). It is similar to how to derive Eq. (13) in [38] from Eq. (6) in [38]. We will fix the typo and
> add the derivation process in the revised paper.
>
> We agree that **from the EBM perspective, our work unifies many previous approaches in a more principled way**. More
> importantly, the unique advantages of EBMs allow us to get much better *compositional generalization* results in
> controllable generation than previous works. We will add this discussion to the revised paper. We also hope the reviewer
> could value the importance of a simple and effective method that is also theoretically plausible. Please note that our
> intention is not to present a simple approach in a complex way. We can move the derivation to the appendix if you
> prefer.
>
> **Q:** There is no ablation regarding a simple Euler discretization method. Is it important to use [5] or does a simple
> Euler discretization work just as well?
>
> We thank the reviewer for the suggestion. In fact, the ODE solver [5] has an option of “euler” that stands for the Euler
> discretization method (see https://github.com/rtqichen/torchdiffeq for details). We used the default adaptive-step
> method (“dopri5”: Runge-Kutta of order 5) because of its adaptivity in step size for better efficiency. As the reviewer
> suggested, we have run our method with the “euler” method (called LACE-euler) on CIFAR-10, with an extra hyperparameter
> “step_size” set to 1e-2 or 1e-3. We can see that 1) **LACE-ODE with the default “dopri5” method is faster than LACE-euler**
> for getting similar performance. 2) **Our method also works decently well with the Euler method**, and its performance lies
> in-between that of LACE-LD and LACE-ODE.
>
> | Methods                       | Inference Time | FID           | ACC          |
> | :---:                         |    :----:      |      :---:    | :---:        |
> | LACE-LD                       | 0.68           | **4.30**      | 0.939        |
> | LACE-euler (step_size=1e-2)   | 0.68           | 6.31          | 0.969        |
> | LACE-euler (step_size=1e-3)   | 6.80           | 5.36          | 0.964        |
> | LACE-ODE                      | **0.50**       | 6.63          | **0.972**    |
>
> **Q:** Why not formulate directly the distribution of x/z conditioned on c instead of talking about the joint
> distribution most of the time when the goal is to sample the former?
>
> When working with energy-based models, the distribution of x/z conditioned on c ($p(x|c)$ or $p(z|c)$) differs from the
> joint distribution ($p(x, c)$ or $p(z, c)$) by a constant factor ($p(c)$) that does not participate in the sampling
> process of the conditional. That’s why when we model the joint energy function, we can easily sample from the
> conditional distribution by simply fixing the conditioning variables in the joint and running the sampling algorithm.
> This does not introduce additional complexity and challenges to our training.
>
> **Q:** What is used for $\beta(t)$?
>
> We gave the form of $\beta(t)$ in Appendix A.2 (line 497), which exactly follows [38]. We will introduce $\beta(t)$ in
> the main text as well in the revised version.
>
> **Q:** How does training time on w space compare to training time on image space?
>
> Training time on image space is more than **20x slower** than w space on a single NVIDIA V100 GPU.
>
> **Final remark**: Thank you for providing constructive and detailed feedback on our submission. If you have any
> additional questions/comments/concerns, please feel free to let us know here. **Otherwise, we would appreciate it if you
> consider raising your rating of this submission.**

---

> > ### Author Response · Authors · 2021-08-24
> > **Follow-up message to Reviewer vR7r**
> >
> > Dear reviewer,
> >
> > You provided very constructive feedback and made a good connection between our work and the prior work on network visualization using generators. As we pointed out in our response, our derivation of the ODE-based sampler sheds light on this connection in a more principled way from the perspective of energy-based models. You also felt that we “hide a simple approach in a complex derivation”. In fact, our intention is quite opposite of that, and we have emphasized the simplicity of our method in the abstract, intro, and the method section (the word *simple* is mentioned 5 times in the paper). We provided the derivation for the sake of completeness, and we are happy to move it (Eq. 7 to 10) to the appendices if you believe it distracts the reader and hides the simplicity of our model.
> >
> > **We are wondering if our response in the previous message could address your concerns and whether you would consider raising your rating for this submission.** We believe that this submission proposes a novel and simple approach for compositional and controllable image generation, and we show that it achieves state-of-the-art results in several tasks, outperforming complex models like StyleFlow. This submission could be of strong interest to the community, and thus we are working hard now on releasing our source code publicly to facilitate research in this space. Please feel free to let us know if you have any additional questions or concerns.

---

> > > ### Comment · Reviewer_vR7r · 2021-09-01
> > > **Thank you.**
> > >
> > > Thank you for your detailed response!
> > >
> > > Yes, I strongly believe that the derivation in Sec. 2.3 together with the overall presentation of the method hides the (positive) simplicity of the approach. I appreciate the commitment to improve the presentation by moving the derivation to the supplementary.
> > >
> > > My other main concern is that the novelty and contribution is repeatedly described based on the fact that the proposed approach does not need to train an EBM in pixel space or use MCMC during training on a latent code. But this is only possible because it is assumed that a GAN was already trained in pixel space and provides a mapping between a low-dimensional, normally distributed space and images. The latter is essentially the main difficulty in generative modeling. Therefore, I find the positioning of this paper as being an energy based model (i.e. a generative model) to be misleading. I am less concerned about the fact that this leads to a two-stage approach which is not optimized end-to-end (as pointed out by joed), but by the fact that here, most of the problem is solved in the first stage and the paper addresses only the second stage. In contrast, works like [Diagnosing and Enhancing VAE Models](https://arxiv.org/abs/1903.05789), [Generative Latent Flow](https://arxiv.org/abs/1905.10485) and [An Acceleration Framework for High Resolution Image Synthesis](https://arxiv.org/abs/1909.03611) among others, also use a two-stage approach but they use a relaxed first stage consisting of a simple autoencoder which is relatively easy to train, and then learn a generative model (in the form of a VAE, Normalizing Flow or GAN) on top of the autoencoder representation. Note that here, the distribution of the latent (autoencoder) representation is far from being a standard normal distribution. I hope that this helps to make it clear why the assumption that the latent code is distributed normally **is** very restrictive---unless one already has a generative model for the data, it is not applicable. This means that the approach is not really significant as a generative model itself.
> > >
> > > Of course, controlling an existing generative model is an important problem but it is a very different one than obtaining a generative model. Thus, I think the presentation of the approach as an energy based model currently can be misleading. The same algorithm is also much more simple to derive as the gradient flow minimizing the energy in Eq. (4) even though it is nice to show that it can also be interpreted as sampling from an appropriately defined energy based model.
> > >
> > > To summarize, I rephrase my original assessment: The paper presents very good results with a very simple method, but the presentation and claims of contributions can be misleading. A thoroughly revised version will be a good candidate for a strong paper.
> > >
> > > I strongly urge the authors to (i) make it very clear that the work addresses the problem of controlling an existing generative model instead of learning a generative model itself, (ii) simplify the derivation, make sure it is correct, and (iii) discuss the connection between the resulting algorithm in Eq. (10), the gradient flow of the energy in Eq. (4), Euler discretizations of these flows and the Langevin Dynamics with vanishing noise term. Based on the response so far, I assume that the authors will incorporate such changes and thus raise my rating from a 5 to a 6.

---

> > > > ### Author Response · Authors · 2021-09-01
> > > > **Responses to Reviewer vR7r’s rebuttal feedback**
> > > >
> > > > We do appreciate the detailed discussion of your main concern around the positioning of this paper as being an energy-based model (EBM) (i.e. a generative model). We understand your concern much better now. We agree with you that in our framework the challenging part of the high-quality generation in data space is handled by a pre-trained generator (such as StyleGAN). We have mentioned this several times in the paper (e.g., lines 43-45, lines 55-56, and lines 321-322), but perhaps we did not emphasize it enough.
> > > >
> > > > As you highlighted, the role of the EBM in our framework is not to provide generation in the data space or improve the generation quality of an existing generator. In fact, the design of our EBM ensures that the marginal distribution over data stays the same as the original pre-trained unconditional generator $p_g(x)$ (see lines 99-100). By doing so, the EBM is tasked to generate latent variables (i.e., latent $z$ in StyleGAN) that will be decoded to images with particular attributes. We hope you agree with us that this EBM is still a conditional generative model __over latent variables__, designed purely for controllability.
> > > >
> > > > To disambiguate any misunderstanding around this and also to address your earlier comments on the derivations, we will take the following actions for the final camera-ready version:
> > > >
> > > > 1. We will clearly emphasize that the high-quality generation is done with a pre-trained GAN generator, and the role of EBM is _not_ to improve generation quality, but to generate latent variables conditioned on given attributes (i.e., provide controllability to an existing generative model).
> > > >
> > > > 2. We will move Eqs. (7-10) to appendices for completeness and we will fix the missing coefficient 2 in Eq. (7).
> > > >
> > > > 3. We will discuss the connection between Eq. (10) and gradient flows, their Euler discretization, and Langevin Dynamics.
> > > >
> > > > We are humbled by your assessment: “The paper presents very good results with a very simple method, but the presentation and claims of contributions can be misleading. A thoroughly revised version will be a good candidate for a strong paper.”
> > > >
> > > > **We would like to assure you that we are committed to applying these changes to the paper. We hope you agree that these changes are minor and can be done easily in the final camera-ready version.**

---

### Official Review · Reviewer_joed · 2021-07-15

**Rating:** 4
**Confidence:** 5

**Summary:**

This paper studies conditional learning by training an EBM in the latent space of a pre-trained top-down generator such as StyleGANs. The EBM is a joint distribution of data and attributes together, and sampling from it is formulated as solving an ordinary differential equation. Experimental results show that the method outperforms the state-of-the-art in both conditional sampling and sequential editing. The contribution of the paper lies on obtaining state-of-the-art performance of conditional learning by combining existing technologies, such as StyleGAN, latent space energy-based learning, and score-based generative model via stochastic differential equations.

**Limitations And Societal Impact:**

Despite the state-of-the-art results, the paper has the following limitations.

(1)	Insufficient novelty. (i) The performance of the proposed methods, i.e., the high quality of the generated images, comes from the pretrained generator of the StyleGANs, and the sampling of the EBM relies on the score-based generative model. Therefore, the results are not surprising because the paper just combines all existing state-of-the-art works together, and such a contribution is very straightforward. Even though the proposed pipeline to train such a conditional model is useful for the application, but the novelty is insufficient for the conference.  (ii) As to the latent space energy-based modeling, the EBM built on the data and attributes actually the posterior distribution of the top-down generator (a latent variable model). Sampling the latent space given the data via Langevin dynamics has been well studied in [1][2][3][4], in which the generator is trained by MLE where sampling the latent space conditioned on the data is required. This step is called MCMC inference. The energy function of the posterior distribution is the equation (6) in the current paper.  That means the latent space energy-based modeling is not new.

[1] Alternating back-propagation for generator network. AAAI-17.

[2]  Learning dynamic generator model by alternating back-propagation through time. AAAI-19

[3] Cooperative learning of descriptor and generator networks. IEEE Transactions on Pattern Analysis and Machine Intelligence (PAMI 2020)

[4] Cooperative Training of Fast Thinking Initializer and Slow Thinking Solver for Conditional Learning. (arXiv 2019 or PAMI 2021)

(2) Insufficient references. (i) The paper lacks some discussion of related works [1][2][3][4], which also samples the latent space of the generator by MCMC or uses the same form of latent space EBM as the current manuscript. (ii) missing pioneering works of ConvNet-EBM. The current successful deep energy-based learning framework was first proposed in Xie et al. 16 [5], where deep neural network is proposed to parameterize the energy function, and MLE with Langevin dynamics is used to train the parameters. The current paper does not cite those published prior arts before 2019, such as [5][6][7][8]. This might lead to negative ethical issues.

[5] A Theory of Generative ConvNet (ICML 2016)

[6] Synthesizing Dynamic Pattern by Spatial-Temporal Generative ConvNet (CVPR 2017)

[7] Learning Descriptor Networks for 3D Shape Synthesis and Analysis. (CVPR 2018)

[8] Learning generative ConvNets via multigrid modeling and sampling. (CVPR 2018)

(3)	Insufficient comparison. The paper might need to compare with [4], which is also related to controlled image generation.

The limitations of the current paper are the insufficient novelty and insufficient references, thus I suggest a rejection.


**Main Review:**

This paper addresses an important problem in computer vision, which is learning conditional generative models for controlled image generation. The paper combines the existing state-of-the-art techniques, including StyleGAN, energy-based model, and score-based generative models, to achieve the state-of-the-art performance of conditional generation. The paper is well-written, well-organized, and easy to follow.

**Time Spent Reviewing:**

48h

---

> ### Author Response · Authors · 2021-08-10
> **Responses to Reviewer joed**
>
> We thank the reviewer for taking the time to review our paper, and we would like to provide a clarification on our
> novelty and a discussion with the provided references.
>
> **Q:** The results are not surprising because the paper just combines all existing state-of-the-art works together, and
> such a contribution is very straightforward.
>
> We respectfully disagree with the reviewer’s assessment that this work is a *straightforward* combination of different
> SOTA methods. Our work is the first to combine the generative power of GANs and compositionality of EBMs to achieve
> high-quality controllable generation of large images (1024x1024 px).
>
> A straightforward combination of joint EBM (i.e. JEM [12]) and StyleGAN is the Latent-JEM baseline (see Appendix A.3)
> that we introduce as a competing baseline. Like previous EBMs (independent of the pixel-space or latent-space EBMs),
> Latent-JEM requires MCMC sampling for training which in practice results in slow and unstable training. However, **our
> joint EBM formulation in the latent space does not perform MCMC sampling *during training***, making it simpler and
> faster while achieving better performance.
>
> To our knowledge, no prior work on EBMs has applied the ODE sampling. The ODE sampler in our method is not a
> straightforward application of score-based models [38], and the difference has been extensively discussed in our work
> (lines 142-144): [38] requires training both the score function and the *time-variant* classifier, making the training
> and inference slow and challenging. By moving to the latent space, however, **our ODE sampling only needs to train a
> *time-invariant* classifier and does not require training the score function**, making the training much simpler and the
> inference much more efficient.
>
> **Q:** The EBM built on the data and attributes is actually the posterior distribution of the top-down generator (a
> latent variable model). The energy function of the posterior distribution is the equation (6) in the current paper.
> Sampling the latent space given the data via Langevin dynamics has been well studied in [1][2][3][4]. That means the
> latent space energy-based modeling is not new.
>
> The formulation here is different from the references suggested by the reviewer. The papers (in particular [1,2])
> suggested by the reviewer are focused on sampling from the posterior distribution $p(z|x)$ (x: data, z: latent variable)
> using the Langevin dynamics (LD) in latent-variable generative models. However, our joint energy function in Eq. (6) is
> obtained by a simple reparameterization trick in implicit GAN generators. Instead of learning the posterior $p(z|x)$ as
> in [1,2], **our method does not require inferring latent variables or sampling from the EBM using MCMC during training**
> . Thus, our formulation is fundamentally different from those in the suggested papers.
>
> Because of the difference in the formulation, the EBMs in the shared papers [1-4] all need to train the generator via
> MCMC sampling, while our EBM defined in Eq. (6) does not require training the generator. **This advantage distinguishes
> our method from previous EBM methods regarding training speed and inference efficiency.** Please note that we will be
> more than happy to cite all the suggested references and discuss differences.
>
> **Q:** The paper lacks some discussion of related works [1][2][3][4]. The current paper does not cite those published
> prior arts before 2019, such as [5][6][7][8].
>
> We have discussed how our method differs from the shared papers [1-4] above, and we are happy to incorporate this
> discussion in the revised paper. Thanks for pointing out early works on EBMs, and we are also happy to add them to the
> related work section.
>
> **Q:** The paper might need to compare with [4], which is also related to controlled image generation.
>
> In [4], the closest experimental setting is the class-conditional generation on CIFAR-10, where the IS score of the
> proposed method is 7.30. However, our method has an IS score of 9.87, showing that **our method achieves much better
> image quality than [4]**. We will be happy to compare our method to the suggested reference.

---

> > ### Comment · Reviewer_joed · 2021-08-20
> > **limited novelty**
> >
> > We thank the authors for their time to reply to the review. The authors did a very good job to clarify some of my concerns and the difference between the references and the current paper under review. It is always better to compare with similar works even if they have small differences. In the reply of the authors, their answers provide a good connection from the prior works as well as the difference between them to better understand which parts of the proposed method are new and how significant the new parts are. I hope the author can involve all the valuable discussion into their revision.
> >
> > However, my major concern is still on the novelty and contribution of the paper:
> > The current latent space modeling mainly has the following two directions: (ii) two-stage / not an end-to-end framework: This class method first extracted low-dimensional features of the high-dimensional training data, and then build an EBM model on the extracted features. (ii) The framework builds a model where an EBM model is on top of the top-down generator. The scheme (i) is less novel than (ii), because (i) can separate the learning into two stages, and each stage can use its own existing techniques. (i) is more straightforward. But scheme (ii) requires the training of two parts in one single objective, which will lead to a more consistent and statistically rigorous estimation of all parameters. And such end-to-end learning of a new hierarchical model usually is harder but more novel and significant than two-stage learning. The current framework belongs to (i), even though building on pre-trained state-of-the-art SyleGAN generator, the subsequent latent space EBM can easily obtain impressive performance, I have to say such a framework doesn’t meet my standard of novelty.
> >
> > For example, [1] (belonging to the scheme (ii)) is a top-down generator with an EBM on top of the latent space, where the latent space also has two parts, which are a continuous latent vector and a symbolic one-hot vector. The whole model is trained end-to-end in a single objective, and the goal is also for a controllable generation. Compared with [1], the current two-stage learning and modeling framework looks less novel, compared with existing works.
> > As to the sampling through an ODE solver, I agree with Reviewer vR7r that using ODE for sampling looks not that natural and might not be necessary. I believe using short-run MCMC like [1] can be more natural and automatic (because it can be derived from the model itself) for sampling in the latent space. MCMC in the latent space is easier to converge than that in the data space. This makes other sampling techniques not that necessary.
> >
> > Therefore, I update my rating (from 3 to 4) based on the feedback and the novelty compared with existing similar works.
> >
> > [1] Latent Space Energy-Based Model of Symbol-Vector Coupling for Text Generation and Classification. ICML 2021

---

> > > ### Author Response · Authors · 2021-08-21
> > > **Further clarification**
> > >
> > > We thank the reviewer for the discussions, and we are pleased to hear that our rebuttal has clarified some concerns.
> > >
> > > Regarding the novelty concern, the reviewer raised two schemes of latent space EBM modeling, where we think the key difference is whether or not we train/fine-tune the generator (such as VAE and GANs) that the latent space EBM has been built on. Namely, many previous works on latent space EBM modeling (including our reference [31] and its follow-up work [1] shared by the reviewer) need to train the generator while our method relies on a pre-trained unconditional generator. We agree with the reviewer that our method is simpler than other methods in training, but we respectfully disagree that the simplicity implies the lack of novelty. Instead, provided that our method outperforms many strong baselines and achieves state-of-the-art results, we value the simplicity of our method as one of our key contributions. Additionally, by formulating the training in two stages, we can pre-train an unconditional (GAN) generator on a large set of unlabeled data, and then, introduce the EBM models for controllability on the fly for each conditioning attribute separately.
> > >
> > > Regarding the correctness/naturalness of the ODE sampling method, Reviewer vR7r had some confusion on the ODE formulation in Eq. (10), we believe this confusion mainly comes from a typo (i.e., a missing constant “2” coefficient) in Eq. (7).  In the rebuttal to the comments of Reviewer vR7r, we have clarified that “the ODE formulation in Eq. (10) is correct after fixing the typo”. Therefore, we would like to bring it to the reviewer’s attention that our ODE sampling for EBMs has also been *naturally* derived from a reverse SDE [38] for controllable generation. In fact, ODE based samplers are now heavily used by continuous-time denoising diffusion models such as [38] and reference [a].
> > >
> > > [a] Vahdat et al. Score-based Generative Modeling in Latent Space, 2021.
> > >
> > > Regarding its necessity, we found that even in the latent space, sampling with LD (or Langevin MCMC) is still quite sensitive to several hyperparameters (such as number of steps, learning rate, scale of noise, etc.) while the black-box ODE solver is more robust to its tolerances. For example, our ablation studies in Figure 5 in Sec. 3.4 and Figure 16 in Appendix A.9 have systematically compared the ODE and LD sampling, and the results showed the advantages of the ODE sampling over LD in terms of both robustness to hyperparameters and inference speed. In summary, we believe that introducing a new effective sampling method to EBMs will add value to the energy-based model community, and will inspire more related sampling methods for EBMs beyond the de facto LD sampling.
> > >
> > > We hope our answers can further address the reviewer’s concerns on the novelty of our work and the necessity of the ODE sampling. If you have any additional questions/comments/concerns, please feel free to let us know here.

---

### Official Review · Reviewer_QDS5 · 2021-07-16

**Rating:** 7
**Confidence:** 2

**Summary:**

Using labelled real data, an EBM is trained in the latent space of a pretrained GAN, rather than directly in pixel space. An ODE solver is used to sample from the EBM rather than Langevin Dynamics, which is shown to be more robust to hyperparameter settings. Using properties of the EBM, conditional generation compositional editing can be demonstrated, including in zero-shot scenarios.

**Limitations And Societal Impact:**

Limitations and broader impact are addressed.

**Main Review:**

The advantages of this approach are that it operates in latent space rather than pixel space, so training and inference is fast. Also, only the EBM portion is trained and synthesis uses a pretrained GAN, so it can combine the compositionality properties of EBMs with the image quality of recent GANs.

A few points of further clarification:

In the energy update function Eqn 1, samples $x$ are drawn from both the real data and the model, however in line 160 it is mentioned that the classifier is trained in the w space of StyleGAN, so it seems like the w-space is used as input to the classifier f, which does not naturally exist for real images. How are the real images provided as inputs to the classifier with respect to Eqn 1?

In the CIFAR10 experiment, the StyleGAN-ada unconditional model FID is reported as 2.92, and conditional as 2.42, whereas in Table 1 the reported FID is slightly higher than that of the pretrained models. What causes this discrepancy, since the same model is used as the image generator?

Overall, the writing is clear and compositional results seems convincing.

**Time Spent Reviewing:**

2.5

---

> ### Author Response · Authors · 2021-08-10
> **Responses to Reviewer QDS5**
>
> We thank the reviewer for the positive assessment, and we clarify some questions that the reviewer raised as follows.
>
> **Q:** How are the real images provided as inputs to the classifier with respect to Eqn 1?
>
> To train the latent classifier in the $w$-space, we generate 10k (image, $w$) pairs by sampling from the pre-trained GAN
> model. We label each $w$ latent by annotating its paired image with an image classifier. Please see the “data
> preparation” part (lines 508-521 in Appendix A.2) for more details of how we prepare data for the latent classifier,
> which has also been used in the StyleFlow paper [1].
>
> **Q:** In Table 1, the reported FID is slightly higher than that of the pretrained models. What causes this discrepancy,
> since the same model is used as the image generator?
>
> Good point! **The sampling process causes this difference in the FID.**
>
> Our goal is to turn an unconditional generative model into a conditional one for better controllable generation. This
> allows us to train an unconditional generator once in a domain and introduce conditioning attributes later. We target
> both the controllability (ACC) and image quality (FID), where a trade-off may exist between these two metrics (Figure 5)
> . However, the conditional StyleGAN-ADA mainly aims for better FID and it may sacrifice the ACC score. We will add this
> discussion to the revised paper.
>
> Regarding the root cause of the FID discrepancy, original StyleGAN-ADA **randomly** samples the latent z (by following a
> standard Gaussian) for image generation, while our method **controllably** samples the latent z (to satisfy the
> conditional attribute specifications) with the ODE/LD sampler. The resulting data distributions of the two sampling
> methods will be different, thus making the FID different.

---

> > ### Comment · Reviewer_QDS5 · 2021-08-26
> > **Reviewer Response**
> >
> > Thanks to the authors for the response. I will retain my 7 rating.
> >
> > Q1: Ok, this is clear to me. Initially I was confused whether you need to train on real images and thus how the corresponding latent code would be obtained, but it seems this is not the case and it is trained entirely on generated images (using an auxiliary classifier to get labels)
> >
> > Q2: Thanks for this discussion.

---

### Decision · Program_Chairs · 2021-09-27

**Decision:**

Accept (Poster)

**Comment:**

The paper proposes an approach for controllable image generation based on applying an energy-based model in the latent space of a pre-trained unconditional generative model. The reviews are mixed: three reviewers are in favor of acceptance, while one recommends rejection. After reading the reviews, the rebuttal, the discussion, and the paper itself, below are some key points.

Pros:
1) The method is sensible and simple
2) The method has been evaluated thoroughly and shown to work well compared to the relevant baselines
3) Quite thorough analysis experiments evaluating different design decisions

Cons:
1) Limited novelty - the work mainly combines existing methods and is related to many prior papers
2) High-res experimental results only on one dataset

Overall, the proposed method is not overwhelmingly new, but it is simple and effective. The paper is well written and easy to understand, and the experimental evaluation is fairly thorough. I thus recommend acceptance at this point, but I urge the authors to take the reviewers' comments into account and adjust the final version of the paper accordingly.